# Galectin-9 regulates the threshold of B cell activation and autoimmunity

Logan K Smith[1,2], Kareem Fawaz[1], Bebhinn Treanor[1,2,3]*

[1]Department of Biological Sciences, University of Toronto Scarborough, Toronto, Canada; [2]Department of Immunology, University of Toronto, Toronto, Canada; [3]Department of Cell and Systems Biology, University of Toronto, Toronto, Canada

**Abstract** Despite the mechanisms of central and peripheral tolerance, the mature B cell compartment contains cells reactive for self-antigen. How these cells are poised not to respond and the mechanisms that restrain B cell responses to low-affinity endogenous antigens are not fully understood. Here, we demonstrate a critical role for the glycan-binding protein galectin-9 in setting the threshold of B cell activation and that loss of this regulatory network is sufficient to drive spontaneous autoimmunity. We further demonstrate a critical role for galectin-9 in restraining not only conventional B-2 B cells, but also innate-like B-1a cells. We show that galectin-9-deficient mice have an expanded population of B-1a cells and increased titers of B-1a-derived autoantibodies. Mechanistically, we demonstrate that galectin-9 regulates BCR and distinct TLR responses in B-1a cells, but not B-1b cells, by regulating the interaction between BCR and TLRs with the regulatory molecules CD5 and CD180, respectively. In the absence of galectin-9, B-1a cells are more readily activated and secrete increased titers of autoantibodies that facilitate autoantigen delivery to the spleen, driving autoimmune responses.

**\*For correspondence:**
bebhinn.treanor@utoronto.ca

## Introduction

B cells contribute to host immunity through their potent capacity to activate T cell responses, modulate inflammatory profiles, produce high-affinity antibodies, and provide lasting immunological memory (*Hoffman et al., 2016*). Many of these functions are initiated by binding antigen to the B cell receptor (BCR), triggering downstream signaling and ultimately shaping cell fate (*Hasler and Zouali, 2001*). BCR signaling, however, must be tightly regulated to prevent aberrant B cell activation to low-affinity self-antigens and development of autoimmunity. Indeed, since the specificity of the BCR is stochastically generated, B cells with specificity for self-antigen arise (*Kurosaki et al., 2010*). Receptor editing, clonal deletion, and anergy limit the generation and activation of autoreactive B cells; yet despite these mechanisms of central and peripheral tolerance, the mature B cell compartment contains cells reactive for self-antigen (*Liu et al., 2010*). How these cells are restrained from responding to low-affinity endogenous antigen, and the mechanisms that regulate antigen affinity discrimination are not fully understood.

BCR signal transduction is regulated by the nanoscale distribution of the BCR with respect to co-stimulatory molecules that antagonize or enhance signal transduction (*Treanor, 2012*). Loss or engagement of co-stimulatory molecules such as CD45, CD19, and CD22 dramatically alter the intensity of BCR signaling and ultimately shape B cell activation (*Hasler and Zouali, 2001*). These regulatory networks play a key role in tuning B cell responses in secondary lymphoid organs and help to restrain inappropriate activation to innocuous antigens (*Goodnow et al., 1990*). In fact, co-stimulatory molecules such as CD19 and CD45 have been shown to alter the threshold of B cell activation, and perturbations in their function can lead to autoimmunity (*Goodnow et al., 1989*). However, the mechanisms that establish the molecular interaction between BCR and positive and negative co-stimulatory molecules are not fully clear.

Galectin-9 (Gal9) is a soluble, bivalent glycan-binding protein that binds directly to IgM-BCR and CD45 (*Cao et al., 2018*; *Giovannone et al., 2018*). Binding of Gal9 alters the organization of the BCR, enhancing interactions with inhibitory co-receptors CD45 and CD22, thereby reducing BCR signal transduction (*Cao et al., 2018*). These findings raise the question of whether Gal9 acts as a rheostat for antigen affinity discrimination, ultimately restraining B cells from responding to low-affinity antigens and therefore limiting autoimmune responses. Moreover, many B cell subsets have unique regulatory networks and play distinct roles in host immunity. For example, B-1a cells, which reside in the peritoneal and pleural cavities, are poised to rapidly respond to foreign antigen by producing natural IgM; yet their BCR polyspecificity and cross-reactivity for self has also implicated them in the pathogenesis of autoimmune diseases (*Wollenberg, 2011*). Whether Gal9 exerts subset-specific roles on B cell function and the importance of these on the pathogenesis of autoimmunity is unknown.

Here, we demonstrate that loss of Gal9 in mice results in a breakdown of peripheral tolerance and leads to spontaneous autoimmunity with age, marked by splenomegaly, spontaneous germinal center formation, autoantibody production, and nephritis. Mechanistically, we demonstrate that Gal9 regulates the threshold of follicular B cell activation, and loss of this regulatory network permits activation to low-affinity and low-density antigens. Additionally, we observe an expansion in the B-1a compartment in the absence of Gal9 and demonstrate that these B-1a cells, but not B-1b cells, are more sensitive to BCR and distinct toll-like receptor (TLR) stimuli. We identify that Gal9 directly regulates BCR and distinct TLRs by binding to IgM-BCR and CD5 on the surface of B-1a cells as well as TLR4 and the regulatory molecule CD180, dampening signal transduction by altering their nanoscale co-distribution. We demonstrate that enhanced activation of B-1a cells in the absence of Gal9 is detrimental and exacerbates autoimmunity by facilitating transfer of autoantigens to secondary lymphoid organs where they drive autoimmune responses.

## Results

### Gal9 regulates the threshold of B cell activation

BCR signal strength upon antigen binding is defined by a host of co-receptors that fine-tune and coordinate signal transduction. Increased signal strength leads to productive B cell activation when antigen affinity or concentrations are limiting (*Zikherman and Lowell, 2017*). Therefore, BCR co-receptors modulate BCR signaling and define the threshold of antigen required for B cell activation (*Tedder et al., 1997*). We have shown that Gal9 regulates BCR signaling by binding directly to IgM-BCR, altering its membrane organization with respect to the inhibitory co-receptors CD45 and CD22, thereby dampening BCR signal transduction (*Cao et al., 2018*). To investigate whether Gal9 impacts the threshold of B cell activation, we first asked if Gal9 regulates the intensity of signal transduction downstream of BCR ligation. To address this, we stimulated primary naïve murine wild-type (WT) and $Lgals9^{-/-}$ (Gal9KO) B cells with defined concentrations of anti-IgM F(ab')$_2$ as surrogate antigen and measured total tyrosine phosphorylation at 5 min by flow cytometry. Modulating BCR signal strength through limiting the concentration of stimulatory antigen leads to a decrease in the magnitude of the signaling response at 5 min, as observed by the decrease in geometric mean fluorescence intensity (gMFI) of total-phosphotyrosine (*Figure 1A,B*). In the absence of Gal9, however, BCR signaling is enhanced at limiting antigen concentrations compared to WT B cells (*Figure 1A,B*).

Following antigen binding and initiation of the BCR signaling cascade, the BCR-antigen complex becomes internalized and brought to major histocompatibility complex class II (MHCII) containing lysosomes, which facilitate antigen processing and subsequent presentation to T cells, a crucial step to achieve full B cell activation (*Avalos and Ploegh, 2014*). To assess whether altered BCR signaling in Gal9-deficient B cells impacts BCR internalization, we stimulated WT and Gal9KO primary murine B cells with defined concentrations of anti-IgM F(ab')$_2$ and measured surface expression of IgM-BCR over 20 min. Following stimulation with high concentrations of anti-IgM F(ab')$_2$, we see a sharp decrease in surface IgM that plateaus by 20 min, whereas stimulation with decreasing concentrations leads to less IgM internalized (*Figure 1C*). Notably, in the context of reduced antigen concentrations, IgM internalization is increased in Gal9KO compared to WT B cells (*Figure 1C*). To quantify the rate of IgM internalization, we plotted the rate of change (k) of the fitted decay curve, which revealed Gal9KO B cells internalize IgM at a faster rate than WT B cells (*Figure 1D*). Furthermore,

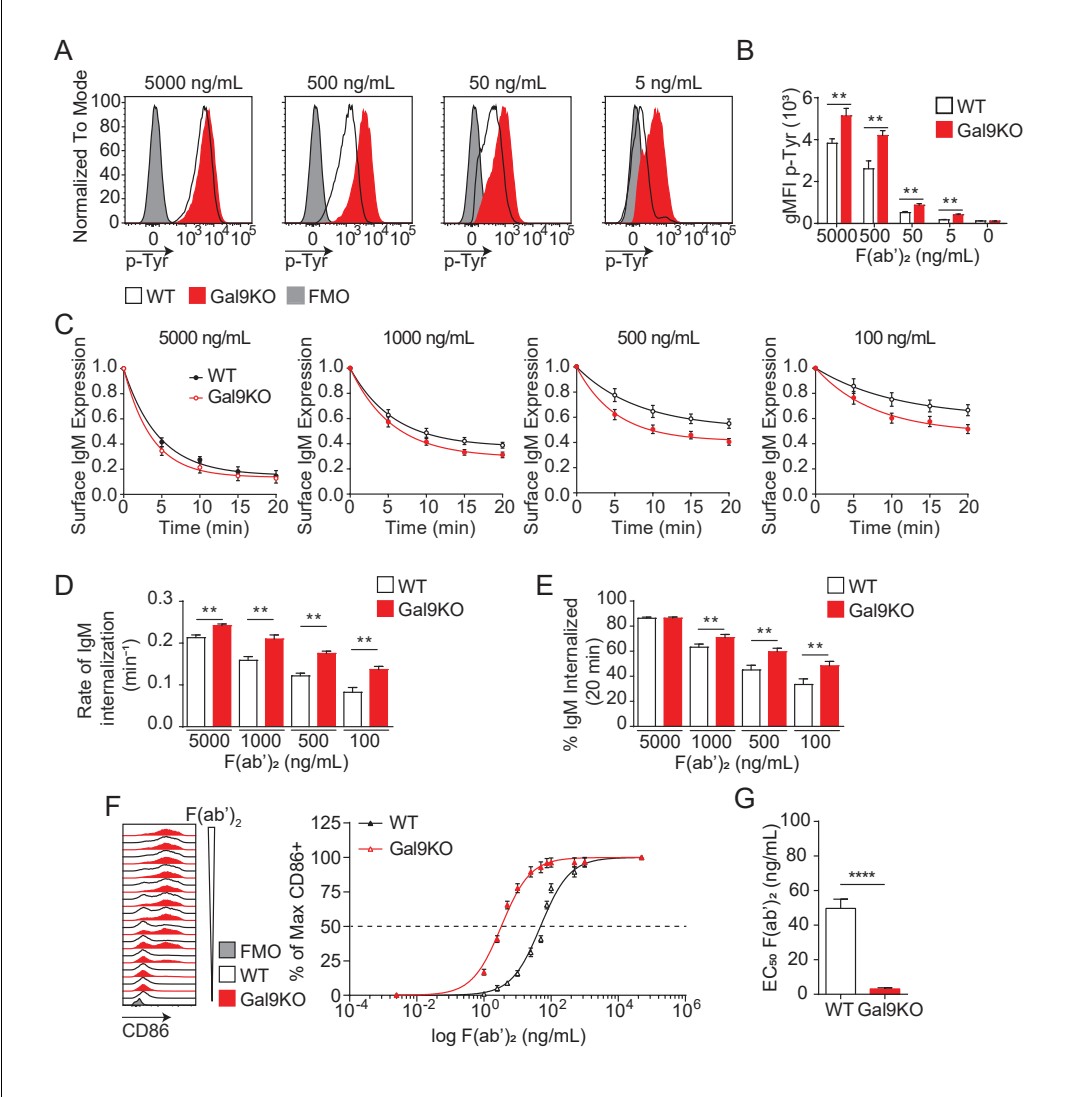

**Figure 1.** Gal9KO B cells respond to limiting concentrations of antigen. (**A**) Representative histograms of total tyrosine phosphorylation (p-Tyr) in WT (black) and Gal9KO (red) B cells at 5 min post-stimulation with anti-IgM F(ab')$_2$, as indicated. FMO (gray shaded). (**B**) Summary geometric mean fluorescence intensity (gMFI) of p-Tyr shown in (**A**). (**C**) IgM internalization over 20 min for WT (black) and Gal9KO (red) B cells following stimulation with anti-IgM F(ab')$_2$ as indicated. (**D**) Rate of IgM internalization (k) of data shown in (**C**). (**E**) Proportion of total IgM internalized at 20 min for data shown in (**C**). (**F**) Representative histograms of CD86 expression on WT (black) and Gal9KO (red) B cells stimulated with increasing concentrations of anti-IgM F(ab')$_2$ (left). Summary statistic, proportion of CD86 expressing B cells (right) as a function of F(ab')$_2$ concentration. (**G**) EC$_{50}$ of F(ab')$_2$ titration shown in (**F**). Data show mean ± SEM and are representative of nine biological replicates over three independent experiments. Statistical significance was assessed by Mann–Whitney **p≤0.01, ****p<0.0001.

The online version of this article includes the following figure supplement(s) for figure 1:

**Figure supplement 1.** Nur77 reporter expression correlates with BCR internalization rate.

**Figure supplement 2.** B cells and macrophage populations are a source of extrinsic Gal9, and are sufficient to alter the threshold of B cell activation.

this results in an increase in total BCR internalization at 20 min to limiting antigen concentrations in Gal9KO B cells (*Figure 1E*). Taken together, these data suggest that increased BCR signaling in Gal9-deficient B cells leads to an increase in internalization of IgM-BCR.

To assess the relationship between BCR signal strength and BCR internalization, we used a Nur77-eGFP reporter system, in which GFP expression is driven by Nur77 and is rapidly upregulated upon BCR signaling (*Zikherman et al., 2012*). Following stimulation for 20 min with defined concentrations of anti-IgM F(ab')$_2$, WT and Gal9KO Nur77-eGFP B cells were washed of excess antigen and allowed to incubate for 16 hr to express GFP proportional to the extent of BCR signal transduction.

Consistent with our previous findings, BCR signal transduction to low-dose stimuli, as measured by the gMFI of the GFP reporter, is enhanced in the absence of Gal9 (*Figure 1—figure supplement 1A,B*). Furthermore, the extent of BCR signaling (gMFI of GFP) correlates with the rate of BCR internalization over the same interval (k, over 20 min) (*Figure 1—figure supplement 1C*). Taken together, these data demonstrate that Gal9 regulates BCR internalization through modulating BCR signal strength.

BCR signal transduction leads to a B cell activation program, marked by upregulation of activation markers and a change in transcriptional and metabolic profiles. We then asked if enhanced BCR signaling in response to limiting antigen concentrations in Gal9KO B cells affects the threshold of B cell activation as assessed by the upregulation of the co-receptor CD86. To address this, we stimulated primary naïve B cells from WT or Gal9 deficient mice with titrated concentrations of anti-IgM F(ab')$_2$ for 16 hr and assessed upregulation of CD86 by flow cytometry. Following 16 hr of stimulation we see increased B cell activation proportional to increasing concentration of anti-IgM; however, in the absence of Gal9 B cells respond to lower concentrations of antigen (*Figure 1F*), quantified by the decreased EC$_{50}$ value (*Figure 1G*). Taken together, these data demonstrate that Gal9 restrains B cell activation when antigen concentrations are limiting.

Gal9 is a pleiotropic protein with many described roles in a variety of immune cells. Because of its secreted nature, Gal9 can be produced by one cell and act on another. This enhanced layer of complexity prompted us to ask which cell types within the spleen can produce Gal9 and contribute to extrinsic sources of Gal9 in the tissue? To address this, we compared Gal9 expression on the surface of a variety of splenic immune cell populations by flow cytometry (*Yu et al., 2016*; *Figure 1—figure supplement 2A*). Additionally, to delineate cells which may be excreting Gal9, we quenched the surface Gal9 with an unlabeled monoclonal antibody, then permeabilized cells, and stained for intracellular Gal9 using a fluorescent conjugation of the same monoclonal antibody (*Figure 1—figure supplement 2B*). Importantly, following quenching with unlabeled antibody, unpermeabilized cells did not stain with fluorescent antibody, demonstrating effective saturation of the quenching antibody in this assay (data not shown). Interestingly, we see high levels of intracellular and extracellular Gal9 expression from many myeloid cell populations, with little detectable Gal9 from the T cell compartment. The contribution of subset-derived Gal9 is a factor of both the expression of Gal9 by a given subset and the overall abundance of that subset within the splenic environment. As such, we normalized the total Gal9 expression (intra- and extracellular) against the relative abundance of each cell subset in the spleen, providing a metric of Gal9 expression that considers the size of specific populations (*Figure 1—figure supplement 2C*). We see that when we factor in subset abundance, B cells appear to have the highest Gal9 expression relative to abundance within the spleen.

We next asked which cell types within the spleen contribute to the Gal9 detected on the surface of B cells. To investigate this, we performed adoptive transfer experiments using Gal9KO B cells transferred into WT hosts, µMT hosts (which lack B cells), and clondronate liposome-treated WT hosts (which results in selective depletion of phagocytic cells, predominantly macrophages/monocytes). We then stained for Gal9 expression on transferred cells to assess the cell type-specific contribution to surface Gal9 expression on B cells. Following treatment with clondronate liposomes, we see approximately 50% reduction in splenic macrophage and monocyte populations compared to control liposome-treated mice (*Figure 1—figure supplement 2D*). Interestingly, when we compare the expression of Gal9 on transferred Gal9KO B cells with endogenous WT B cells in the same host, we see full reconstitution of Gal9 expression on follicular B cells when transferred into WT hosts (*Figure 1—figure supplement 2E*). Notably, however, Gal9 expression is only partially restored when gating on marginal zone B cells (*Figure 1—figure supplement 2F*), suggesting that B cells may have subset-specific requirements for intrinsic vs extrinsic sources of Gal9. Additionally, Gal9 expression was only partially restored when Gal9KO B cells were transferred into macrophage depleted or B cell-deficient hosts (*Figure 1—figure supplement 2E,F*), demonstrating that both B cell-derived and macrophage-derived Gal9 contribute to extrinsic sources of Gal9 within the splenic environment. Lastly, we asked if these perturbations in Gal9 expression affect the threshold of B cell activation. To address this, we stimulated cells with titrated concentrations of anti-IgM F(ab')$_2$ for 16 hr and assessed upregulation of CD86 by flow cytometry. With reconstitution of Gal9 expression by transfer into WT hosts, we see that Gal9KO B cells have similar threshold of activation compared to endogenous WT cells in the same host (*Figure 1—figure supplement 2G,H*). In contrast, we see a decrease in the threshold of activation of Gal9KO B cells transferred into macrophage-depleted or B cell-

deficient hosts, which is proportional to Gal9 expression. Taken together, these data demonstrate that macrophage populations and B cells act as a source of Gal9 within the spleen and that these sources of extrinsic Gal9 are sufficient to alter the threshold of B cell activation.

## Gal9 regulates antigen affinity discrimination

B cell activation is defined by factors such as the abundance of antigen as well as the affinity of that antigen for cognate BCR (*Zikherman and Lowell, 2017*). To assess the role of Gal9 in modulating B cell activation to antigens with reduced affinity, we used the transgenic MD4 system. In this model, B cells express an IgM-BCR specific for model antigen hen egg lysozyme, HEL (HyHEL10, $K_a$ = 4.5 × $10^{10}$ $M^{-1}$). MD4 B cells bind to lysozymes from other avian species with reduced affinity, such as Bobwhite Quail (QEL, $K_a$ = 3 × $10^8$ $M^{-1}$) and Duck (DEL, $K_a$ = 1.7 × $10^7$ $M^{-1}$) (*Lavoie et al., 1992*). To investigate a role for Gal9 in antigen affinity discrimination, we first settled WT or Gal9KO MD4 B cells on artificial planar lipid bilayers displaying HEL, QEL, or DEL antigens, allowing us to simulate antigen presentation in a similar fashion to how B cells interact with antigen in vivo (*Harwood and Batista, 2008*). At 90 s post-interaction, the point of maximal B cell spreading, cells were fixed and imaged using total internal reflection fluorescence (TIRF) microscopy (*Figure 2A*). Gal9KO B cells appear to spread more and accumulate more antigen in response to lower affinity antigens compared to WT B cells. To quantify this, we measured the contact area and total amount of antigen accumulated (*Figure 2B,C*). Indeed, Gal9KO B cells spread more and accumulate more antigen in response to low-affinity antigens presented on lipid bilayers compared to WT B cells. It is well established that the B cell spreading response, and thus the amount of antigen accumulated is regulated by BCR signaling (*Fleire et al., 2006*; *Weber et al., 2008*). Thus, to assess the impact of Gal9 on BCR signaling in the context of an antigen presenting cell, we labeled OP9 stromal cells and deposited lysozyme containing immune complexes onto their surface at three different densities, covering a fourfold range, corresponding to low, medium, and high density. OP9 stromal cells were incubated with WT or Gal9KO MD4 B cells for 5 min, and cell conjugates were then permeabilized, stained for total tyrosine phosphorylation, and analyzed by flow cytometry. In the absence of Gal9, BCR signaling is increased in response to low-affinity and low-density membrane-bound antigens compared to WT B cells (*Figure 2D,E*).

We then asked if this altered signaling in response to low-affinity and low-density membrane-bound antigens in Gal9KO B cells impacts the threshold of downstream B cell activation. To investigate this, we stimulated WT or Gal9KO MD4 B cells with titrated concentrations of these antigens and assessed B cell activation by upregulation of CD86. In the absence of Gal9, upregulation of CD86 is increased in response to lower concentrations of antigen in all three affinities of lysozyme (*Figure 2F*). Furthermore, in the absence of Gal9, B cells internalize IgM at a faster rate in response to low-affinity antigens compared to WT B cells (*Figure 2G*, *Figure 2—figure supplement 1A*). This results in a greater amount of IgM and antigen internalized in Gal9KO B cells when stimulated with low-affinity antigens (*Figure 2H,I*, *Figure 2—figure supplement 1B*). Taken together, these data demonstrate that Gal9 regulates B cell responsiveness to low-affinity and low-density antigens.

## Gal9 regulates autoimmunity

BCR signaling is tightly regulated to mitigate inappropriate activation to otherwise innocuous autoantigens. Many central and peripheral tolerance mechanisms are in place to eliminate B cells with high affinity for autoantigens from the B cell repertoire, resulting in a pool of B cells that should have little to no affinity toward autoantigens (*Basten and Silveira, 2010*). Given our observation of enhanced B cell activation to low-affinity antigens in Gal9-deficient B cells, we then asked, does loss of Gal9 result in spontaneous autoreactivity and the development of autoimmunity in mice? To investigate this, we examined aged (>8 months) WT and Gal9KO mice for evidence of spontaneous autoimmunity. We find that aged Gal9KO mice have enlarged spleens (*Figure 3A*), with detectable germinal center (GC) B cells in the absence of immunization, suggesting they are driven toward autoantigens (*Figure 3B,C*). Furthermore, we see enhanced antibody secreting cell (ASC; defined as FSC$^{hi}$, surface IgM$^{neg}$, and CD138$^{hi}$ expressing cells) (*Pracht et al., 2017*) development in the spleen of Gal9KO mice compared with WT littermate controls (*Figure 3D,E*). Further characterization of the ASC compartment revealed a clear skewing toward B220-negative long-lived ASCs in the absence of Gal9 (*Pracht et al., 2017*; *Figure 3F*). Consistent with this, we observe increased frequency of

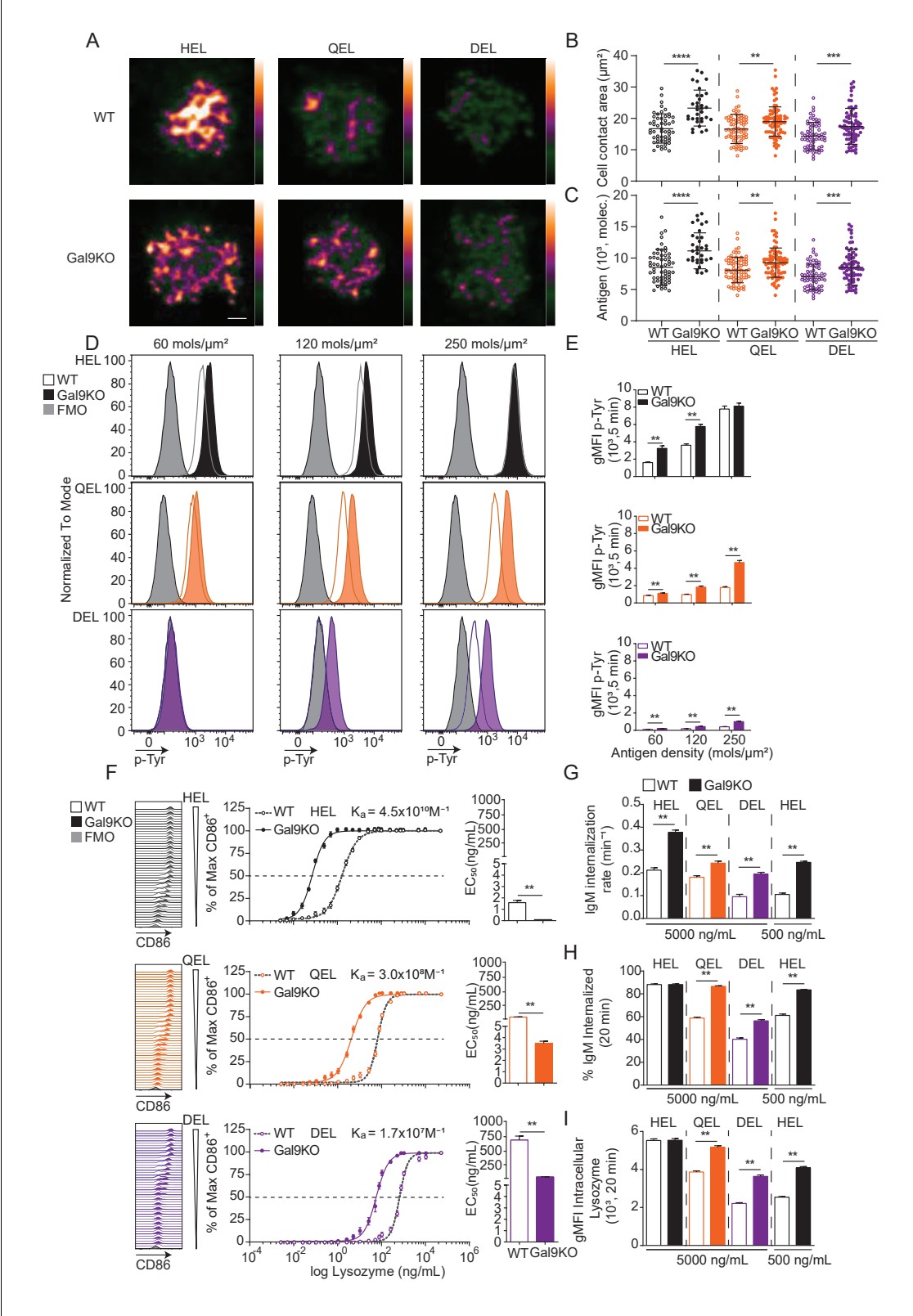

**Figure 2.** Gal9KO B cells respond more readily to low-affinity antigens. (A) Representative images of primary WT and Gal9KO B cells fixed on planar lipid bilayers containing fluorescently conjugated antigen, as indicated, after 90 s of spreading and imaged by TIRF microscopy. Images mapped to a blue-orange ice 8-bit color scale (ImageJ). Scale bar 2 μm. (B) Quantification of cell contact area of WT (open) and Gal9KO (filled) in response to planar lipid bilayers containing HEL (black), QEL (orange), and DEL (purple) antigens. (C) Quantification of the amount of accumulated antigen as in (B). (D)

*Figure 2 continued on next page*

*Figure 2 continued*

Representative histograms of total tyrosine phosphorylation (p-Tyr) in WT (open) and Gal9KO (filled) B cells stimulated for 5 min with indicated antigen deposited on OP9 stromal cells, as indicated. FMO (gray shaded). (E) Summary gMFI of data shown in (D). (F) Representative histograms of CD86 expression on WT (open) and Gal9KO (filled) B cells stimulated with increasing concentrations of lysozymes, as indicated (left). Summary statistic, proportion of CD86 expressing B cells (middle). $EC_{50}$ of lysozyme titration (right). (G) Internalization rate (k) of IgM; (H) Proportion of total IgM internalized; and (I) gMFI of intracellular lysozyme expression for WT (open) and Gal9KO (filled) B cells following 20 min stimulation with lysozyme, as indicated. Data show mean ± SEM and are representative of nine biological replicates over three independent experiments. Statistical significance was assessed by Mann–Whitney **$p \leq 0.01$, ***$p \leq 0.001$, **** $p < 0.0001$.

The online version of this article includes the following figure supplement(s) for figure 2:

**Figure supplement 1.** Gal9KO B cells have enhanced internalization of low-affinity antigens.

B220- ASCs in the bone marrow the absence of Gal9 (*Figure 3—figure supplement 1A,B*). We next asked if these spontaneous germinal centers and increased ASCs are driving autoimmunity. To address this, we looked for circulating autoantibodies in the serum of aged Gal9KO mice. We see increased autoreactivity to HEp-2 cells with serum from Gal9KO mice compared to WT controls, having circulating IgG antibodies that are largely nuclear-reactive and circulating IgM that are largely cytoplasmically reactive (*Figure 3G,H*). Furthermore, antigen-specific anti-dsDNA and anti-GME (glomerular membrane extract) autoantibodies are increased in Gal9KO mice (*Figure 3I,J*). In lupus-like disease, autoantigen containing immune complexes (ICs) are a source of inflammation as they deposit in the kidneys and other organs and impair filtration and kidney function. Thus, we next assessed circulating immune complexes and kidney function in aged WT and Gal9KO mice. We observe an increase in circulating ICs in the serum (*Figure 3K*) as well as increased proteinuria (*Figure 3L*), indicative of nephritis, in aged Gal9KO mice compared to WT controls. Taken together, these data indicate that Gal9KO mice develop spontaneous autoimmunity with age.

T follicular helper (Tfh) cells are essential for the development of autoimmune disease as they are required for the formation and maintenance of GCs (*Stebegg et al., 2018*). A population of regulatory Tfh cells expressing FoxP3 (termed Tfr cells) have been described, and these cells play a critical role in regulating B cell responses in peripheral tissues. Loss of Tfr cells leads to activation of autoreactive B cells and unchecked autoimmunity (*Wollenberg, 2011*). Thus, we measured the frequency of Tfh and Tfr cells in the spleen of aged Gal9KO mice compared to WT controls. We see that aged Gal9KO mice have an increase in Tfh cell frequency (*Figure 3—figure supplement 1C and D*), mirroring the increase in GC B cells (*Figure 3B,C*). Additionally, we observe a similar increase in the frequency of FoxP3+ Tfr cells in the spleen of Gal9KO mice; however, the ratio of regulatory Tfr cells to Tfh cells is similar in both WT and Gal9KO mice (*Figure 3—figure supplement 1E–G*). These data suggest that loss of Gal9 does not lead to spontaneous autoimmunity through impaired generation of regulatory Tfh cells.

BCR signal strength and T cell help are key regulators of tolerance in peripheral tissues. To assess whether there is a breakdown in peripheral tolerance in Gal9KO B cells, we crossed Gal9KO mice onto the MD4/ML5 mouse model (*Goodnow, 2009*). In this system, B cells express a transgenic BCR specific for HEL, as well as soluble HEL secreted by somatic cells. Upon binding antigen in this context, B cells undergo a process of peripheral tolerance called anergy where they become refractory to BCR stimulation, downregulate surface IgM, and have a decreased half-life (*Goodnow et al., 1989*). Therefore, in this model of anergy, there is little to no secretion of antibodies toward the pseudo-autoantigen HEL. We observed a distinct population of cells in the spleen of Gal9KO MD4/ML5 mice that retain high levels of surface IgM (*Figure 3—figure supplement 1H and I*). Furthermore, following IgM stimulation, this population induces robust signaling compared to both anergic WT cells and IgM-low anergic-like cells in Gal9KO mice (*Figure 3—figure supplement 1J,K*). Additionally, we detect higher titers of HEL-specific antibodies in the serum of Gal9KO MD4/ML5 mice compared to MD4/ML5 WT controls (*Figure 3—figure supplement 1L*). These data suggest there is a breakdown in peripheral tolerance in the absence of Gal9, leading to a population of cells that escape tolerance mechanisms.

## Gal9 regulates B-1a cell accumulation

B-1a cells are a population of B cells derived from fetal progenitors that undergo a positive selection process during development, providing B-1a cells with an autoreactive repertoire. It is thought that

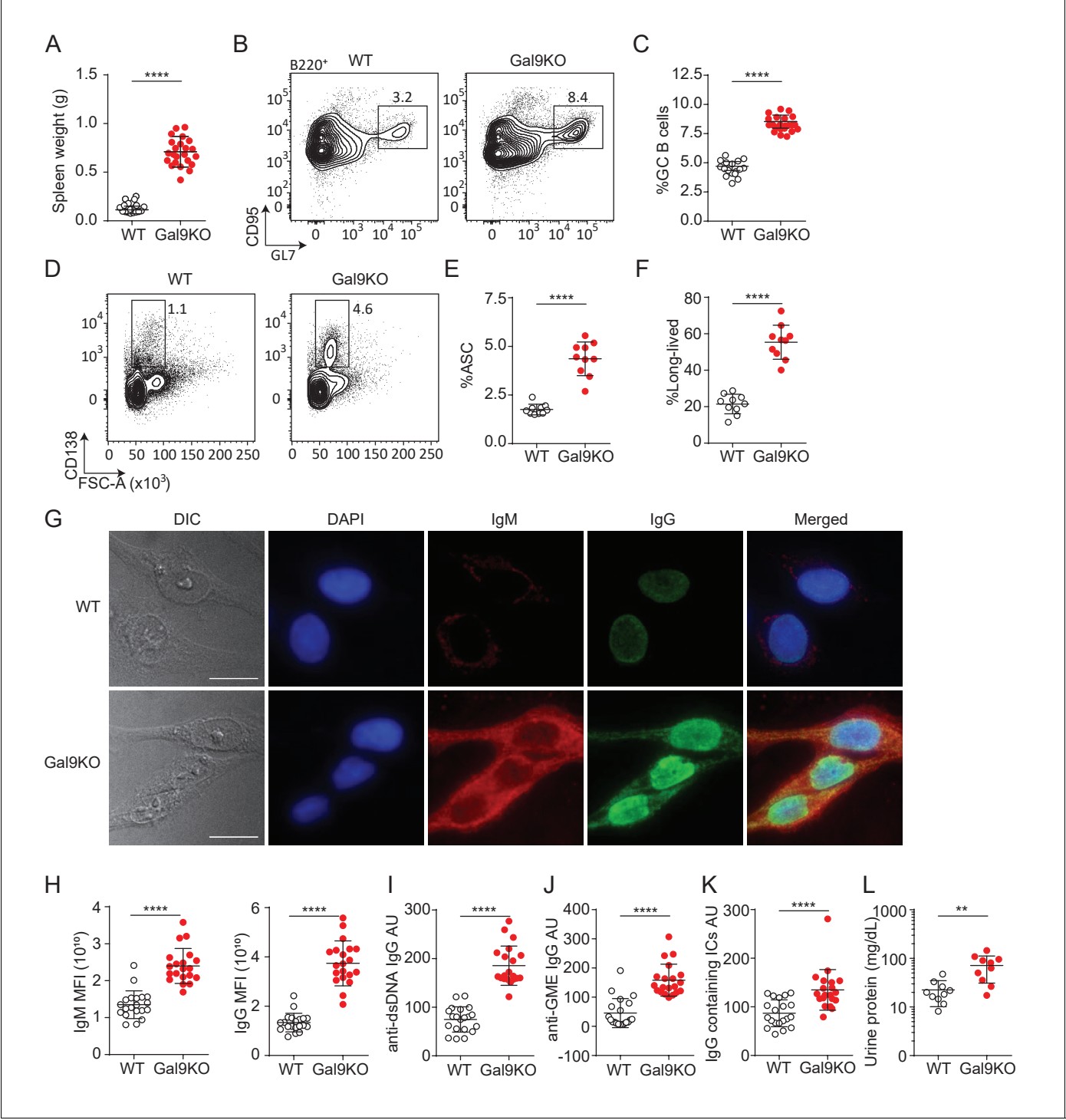

**Figure 3.** Aged Gal9KO mice develop spontaneous autoimmunity. (**A**) Spleen weight of WT (black) or Gal9KO (red) mice aged >8 months. (**B**) Representative plots of GC B cells in the spleen of aged mice, as indicated. (**C**) Summary proportion of GC B cells, as in (**B**). (**D**) Representative plots of antibody secreting cells (ASC) in the spleen of aged mice. (**E**) Summary proportion of ASC, as in (**D**). (**F**) Proportion of ASC lacking B220 expression. (**G**) Representative images of HEp-2 cells stained with sera from WT (top) or Gal9KO (bottom) mice. (**H**) Summary mean fluorescence intensity (MFI) of IgM (left) and IgG (right) of HEp-2 staining as in (**G**). (**I**) Anti-dsDNA specific IgG titers. (**J**) nti-glomerulus membrane extract (GME) specific IgG titers. (**K**) IgG-containing immune complexes (IC) titers. (**L**) Proteinuria, determined by protein secretion into the urine. Data are representative of 10–20 biological replicates over at least three independent experiments. Statistical significance was assessed by Mann–Whitney **p≤0.01, ****p<0.0001.

The online version of this article includes the following figure supplement(s) for figure 3:

*Figure 3 continued on next page*

*Figure 3 continued*

**Figure supplement 1.** Gal9KO B cells can escape anergy.

this is beneficial for aiding in the clearance of cellular debris and senescent red blood cells (*Duan and Morel, 2006*). B-1a cells, defined by their expression of the T cell regulatory protein CD5, along with CD5-negative B-1b cells are typically found at barrier sites such as the lining of the gastrointestinal tract and peritoneal (PerC)/pleural cavities. B-1 cells rapidly respond to both BCR and TLR stimulation and can readily secrete antibodies in the absence of T cell help. Intriguingly, B-1a cells are expanded in several models of autoimmunity, though their specific role remains incompletely defined (*Duan and Morel, 2006*). We therefore asked if B-1a cells were similarly expanded in aged Gal9KO mice. Indeed, we find an increase in the proportion of B-1a cells in the PerC of Gal9KO mice compared to WT mice (*Figure 4A,B*). Additionally, we see an increase in cell number for B-1a cells in the PerC of Gal9KO mice (data not shown). Furthermore, B-1a cells from Gal9KO

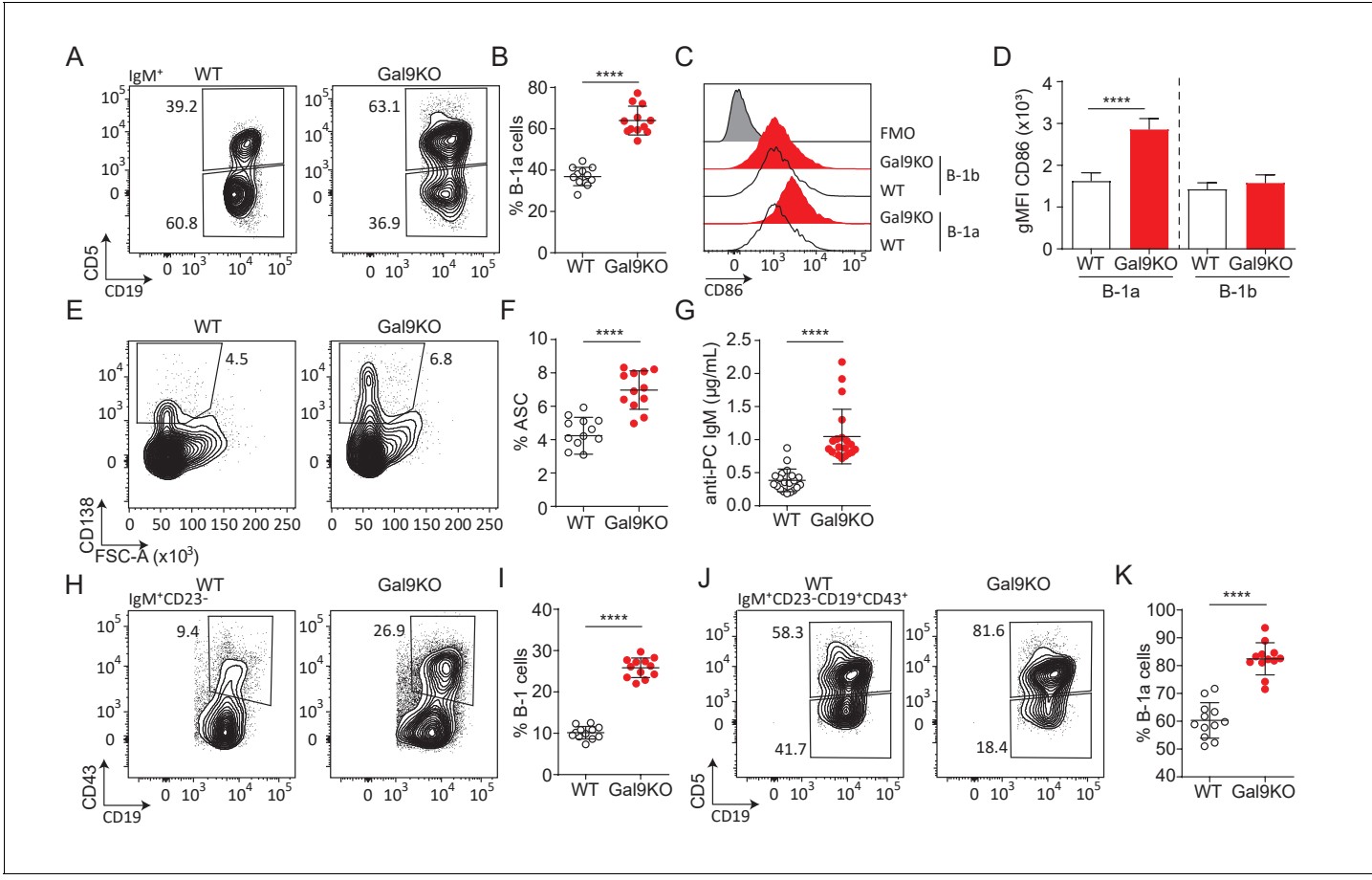

**Figure 4.** Loss of Gal9 leads to enhanced activation and accumulation of B-1a cells at steady-state. (**A**) Representative plots of IgM expressing cells in the PerC of aged WT and Gal9KO mice, as indicated. (**B**) Summary proportion of CD5 expressing B-1a cells in the PerC shown in A. (**C**) Representative histograms of CD86 expression at steady state on PerC cells shown in (**A**). (**D**) Summary gMFI of CD86 expression shown in C. (**E**) Representative plots of ASCs in the PerC of aged mice. (**F**) Summary proportion of ASCs shown in (**E**). (**G**) Anti-phosphorylcholine (PC) specific IgM in sera of aged mice. (**H**) Representative plots of B-1 cells of IgM expressing cells in the spleen of aged mice. (**I**) Proportion of B-1 cells shown in H. (**J**) Representative plots of B-1a and B-1b cells of H. (**K**) Summary proportion of CD5 expressing B-1a cells in the spleen of aged mice shown in (**I**). Data are representative of ten biological replicates over three independent experiments. Statistical significance was assessed by Mann–Whitney ****p<0.0001.

The online version of this article includes the following figure supplement(s) for figure 4:

**Figure supplement 1.** B1–a cells from Gal9KO mice are activated at steady-state.

mice have higher expression of activation markers at steady-state (*Figure 4C,D*; *Figure 4—figure supplement 1A–D*), suggesting that Gal9 restrains B-1a activation.

Activation of B-1 cells readily drives development into ASCs. Consistent with our previous results, we find an expanded population of ASCs in the PerC of aged Gal9KO mice compared to WT littermate controls (*Figure 4E,F*). We then asked if these ASCs were contributing to circulating antibody titers in the serum. A large proportion of B-1a cells share a common specificity for phosphoryl-choline (PC) (*Baumgarth, 2016*). We performed PC-specific ELISAs and found that Gal9KO mice have an increase in PC-specific antibodies, consistent with their increased ASCs (*Figure 4G*). Upon activation at barrier sites, B-1 cells migrate to secondary lymphoid organs (*Ha et al., 2006*). Consistent with this, we see an expanded population of B-1a cells in the spleen of Gal9KO mice compared to WT littermates (*Figure 4H–K*). Additionally, in the NZB/W F1 model of lupus-like autoimmunity, B-1a cells migrate to the thymus and mediate selection of autoreactive T cell clones (*Sato et al., 2004*). Similarly, we see an increase in B-1a cells in the thymus of aged Gal9KO mice (*Figure 4—figure supplement 1E–G*). Taken together, these data demonstrate that Gal9 regulates B-1a cell accumulation and activation at steady-state and implicate B-1a cells in the autoimmune phenotype observed in Gal9-deficient mice.

## Gal9 regulates BCR signaling in B-1a cells

We next asked if the expansion of B-1a cells and their increased expression of activation markers at steady-state in Gal9KO mice is due to altered BCR signaling. To address this, we stimulated PerC B-1 cells with titrated concentrations of anti-IgM F(ab')$_2$ for 16 hr and measured B cell activation by upregulation of CD86. We find that loss of Gal9 leads to enhanced activation of B-1a cells in response to lower concentrations of agonist (*Figure 5A–C*). Notably, this increased activation upon loss of Gal9 is specific to B-1a cells, as we did not observe any difference in B-1b cell responses (*Figure 5—figure supplement 1A–C*). We then asked if Gal9 interactions directly modulate BCR signal transduction in B-1a cells. We stimulated PerC B-1 cells with anti-IgM F(ab')$_2$ for 5 min, followed by fixation and staining for phospho-signaling molecules by flow cytometry. We see that Gal9KO B-1a cells have enhanced phosphorylation of signaling machinery following IgM stimulation (*Figure 5D*). To assess whether enhanced activation is specifically due to Gal9 at the cell surface, we treated WT PerC cells with the pan-galectin inhibitor lactose to remove all galectins from the cell surface and, alternatively, treated Gal9KO PerC cells with recombinant Gal9 (rGal9) to reconstitute Gal9 expression (*Figure 5—figure supplement 2*). Following IgM stimulation, pan-galectin inhibition in WT B cells results in increased signal transduction and importantly, reconstitution of Gal9 in Gal9KO cells directly inhibits signaling in B-1a cells (*Figure 5D*). In contrast, but consistent with our earlier observations, these treatments have little effect on B-1b cell signaling (*Figure 5—figure supplement 1D*). We wondered if differences in Gal9 expression may explain some of the subset-specific discrepancies in B cell activation. Indeed, B-1a cells have higher expression of Gal9 at steady-state compared to B-1b cells (*Figure 5E*). Taken together, our results demonstrate a specific role for Gal9 in the regulation of B-1a, but not B-1b cells.

We have shown that Gal9 regulates B-2 B cell activation through mediating interaction between IgM-BCR and negative co-receptors CD22 and CD45 (*Cao et al., 2018*); however, B-1a cells are known to express a unique surface profile lacking these negative regulators. Therefore, Gal9 regulation of B-1a cells must occur through a unique mechanism. To investigate this, we performed pulldown assays to identify Gal9 ligands in B-1a B cells by incubating protein lysates from sorted B-1a and B-1b PerC B cells or conventional splenic B-2 cells with rGal9-coated beads. We then washed and stained these beads with antibodies targeting various co-receptors and assessed protein enrichment by flow cytometry. We find that Gal9 interacts with IgM-BCR in B-1a, B-1b, and B-2 B cells, but uniquely interacts with the inhibitory glyco-protein CD5 on the surface of B-1a cells (*Figure 5F*, *Figure 5—figure supplement 1E*). Interestingly, we observed only slight enrichment of other known B-1 cell regulators such as Siglec-G or FcγRIIB on Gal9 coated beads, suggesting that Gal9 preferentially interacts with other protein species such as CD5 (*Figure 5—figure supplement 3*). Taken together, these data demonstrate that Gal9 binds to B-1a cells and regulates their activation, likely through the inhibitory co-receptor CD5.

The nanoscale organization of molecules on the cell surface allow for interactions between receptors and important regulatory co-receptors that enhance or restrain signal transduction (*Treanor, 2012*). We have shown previously that Gal9 modulates BCR responses in B-2 B cells by altering

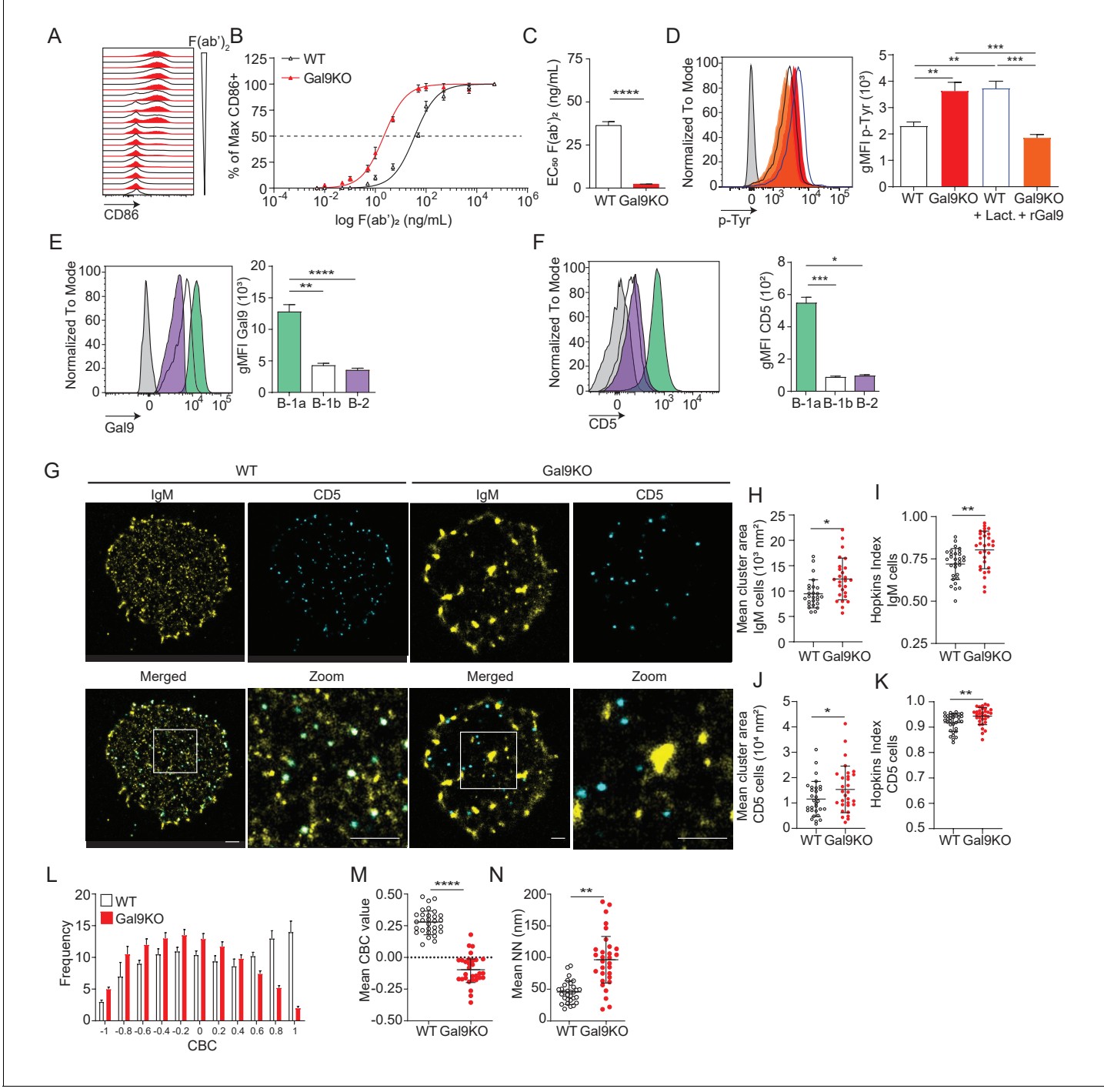

**Figure 5.** Gal9 regulates BCR signal transduction in B-1a cells through interactions with IgM and the negative co-receptor CD5. (**A**) Representative histograms of CD86 expression on B-1a cells from WT (black) and Gal9KO (red) mice stimulated with titrated concentrations of anti-IgM F(ab')$_2$. (**B**) Summary proportion of CD86 expressing cells as a function of F(ab')$_2$ concentration as in (**A**). (**C**) EC$_{50}$ of F(ab')$_2$ titration shown in (**A**). (**D**) Representative histogram of p-Tyr level in B-1a cells from WT (black, open) or Gal9KO (red, filled), and WT B-1a cells treated with lactose (blue, open) and Gal9KO B-1a cells treated with rGal9 (orange, filled) stimulated with anti-IgM (Fab')$_2$ for 5 min (left), summary gMFI of p-Tyr levels (right). (**E**) Representative histograms of Gal9 expression in B-1a cells (green), B-1b cells (white), and splenic B-2 cells (purple) from WT mice (left), summary gMFI of Gal9 expression (right). (**F**) Representative histograms of CD5 detection on rGal9-coated beads incubated with lysates from different B cell subsets, as indicated (left), summary gMFI of CD5 enrichment on beads incubated with lysates (right). (**G**) Reconstructed dual-dSTORM images of IgM (yellow) and CD5 (cyan) on the surface of B-1a cells from WT and Gal9KO mice, ROI (3 μm x 3 μm) used for analysis is expanded in zoom. Scale bars represent 1 μm. (**H**) Quantification of IgM mean cluster area from dSTORM images. (**I**) Quantification of IgM clustering tendency using Hopkins index. (**J**) Quantification

*Figure 5 continued on next page*

**Figure 5 continued**

of CD5 mean cluster area from dSTORM images. (K) Quantification of CD5 clustering tendency using Hopkins index. (L) Frequency distribution of coordinate-based colocalization (CBC) between IgM and CD5. (M) Mean CBC value per ROI. (N) Mean distance between nearest neighbors. Data in (A–F) are representative of nine biological replicates over three independent experiments. Data in (G–N) are representative of 30 ROIs acquired over at least three independent experiments. Data represent mean ± SEM. Statistical significance for C, H, I, J, K, M, and N was assessed by Mann–Whitney, statistical significance for D, E, and F was assessed by Kruskal–Wallis. *p≤0.05, **p≤0.01, ***p≤0.001, ****p<0.0001.

The online version of this article includes the following figure supplement(s) for figure 5:

**Figure supplement 1.** Gal9 does not alter BCR responses in B-1b cells.

**Figure supplement 2.** Lactose treatment inhibits Gal9 binding to B-1 cells, and treatment with recombinant Gal9 restores Gal9 expression.

**Figure supplement 3.** Gal9 binds only marginally to negative co-receptors Siglec G and FcγRIIB in B1 B cells.

the nano-scale distribution of the BCR and its association with inhibitory co-receptors (*Cao et al., 2018*; *Wasim et al., 2019*). To assess the impact of Gal9 on the nanoscale organization of IgM-BCR and inhibitory co-receptor CD5, we performed dual-color direct stochastic optical reconstruction microscopy (dSTORM) to achieve single-molecule resolution of the cell surface. Upon visual inspection of dSTORM images, IgM appears to be more clustered on B-1a cells in the absence of Gal9 (*Figure 5G*). Quantification of cluster area demonstrated that IgM clusters are larger in the absence of Gal9 (*Figure 5H*). To quantify clustering tendency, we calculated the Hopkins index, which assesses clustering tendency relative to a random distribution (0.5). We find that IgM on the surface of Gal9KO B-1a cells has a higher clustering tendency compared to WT B-1a cells (*Figure 5I*). Upon visual inspection of dSTORM images of CD5, we note that it is more highly clustered compared to IgM; however, we do not observe striking differences between WT and Gal9KO B-1a cells. However, quantification of CD5 cluster area and degree of clustering revealed an increase in Gal9KO B-1a cells (*Figure 5J,K*). We hypothesized that Gal9 crosslinks IgM and CD5 to regulate BCR signaling in B-1a cells. Dual-dSTORM provides the advantage of single-molecule coordinate-based colocalization analysis to examine whether there are differences in the colocalization of these molecules on the surface of B-1a cells in the absence of Gal9. Visually, we can discern a loss of colocalized IgM and CD5 clusters in the absence of Gal9 (*Figure 5G*). To quantify this, we performed coordinate-based colocalization (CBC) analysis, which ranges from −1 (perfectly segregated) to +1 (perfectly colocalized). The frequency distribution of CBC values shifts to the left (i.e. toward negative values) in the absence of Gal9, indicating an increase in the segregation of IgM-BCR and CD5, compared to WT B-1a cells (*Figure 5L*). This results in a decrease in the mean CBC value per cell (*Figure 5M*) and a corresponding increase in the mean nearest neighbor (NN; *Figure 5N*). In contrast, but consistent with the lack of effect of Gal9 deficiency on B-1b B cell activation, we find no difference in the organization of IgM on the surface of B-1b cells in the absence of Gal9 (*Figure 5—figure supplement 1F–G*). Taken together, these findings identify a unique and specific mechanism of Gal9 regulation of B-1a BCR signaling.

## Gal9 regulates TLR4 responses in B-1a cells

B-1 cells respond to T cell-independent antigens at barrier sites and rapidly differentiate into effector cells such as ASCs through BCR or TLR stimulation (*Baumgarth, 2016*). To assess whether Gal9 influences innate-like responses of B-1 cells, and to identify TLRs potentially regulated by Gal9, we stimulated PerC cells from WT or Gal9KO mice with titrated concentrations of various TLR ligands and measured the proportion of cells activated by upregulation of CD86. Interestingly, we find that B-1a cells from Gal9KO mice are more sensitive to LPS and CpG stimulation, but not imiquimod or zymosan stimulation (*Figure 6A–D*). In contrast, but consistent with BCR stimulation, Gal9 appears to be dispensable for TLR activation of B-1b cells (*Figure 6—figure supplement 1A–D*). We then asked if enhanced CD86 upregulation in the absence of Gal9 was due to a similar enhancement of TLR signaling in B-1a cells. To address this, we stimulated WT or Gal9KO PerC cells with LPS and assessed total-tyrosine phosphorylation at 10 min by phospho-flow cytometry. We find that Gal9KO B-1a cells, but not B-1b cells, have increased total tyrosine phosphorylation at 10 min post-LPS stimulation, which can be reduced to levels comparable to WT cells by pre-treatment with rGal9 (*Figure 6E*, *Figure 6—figure supplement 1E*). Similarly, pre-treatment of WT B-1a, but not B-1b, cells with lactose to displace galectins, enhanced total tyrosine phosphorylation upon LPS stimulation (*Figure 6E*, *Figure 6—figure supplement 1E*). To identify Gal9 ligands in B-1a cells that

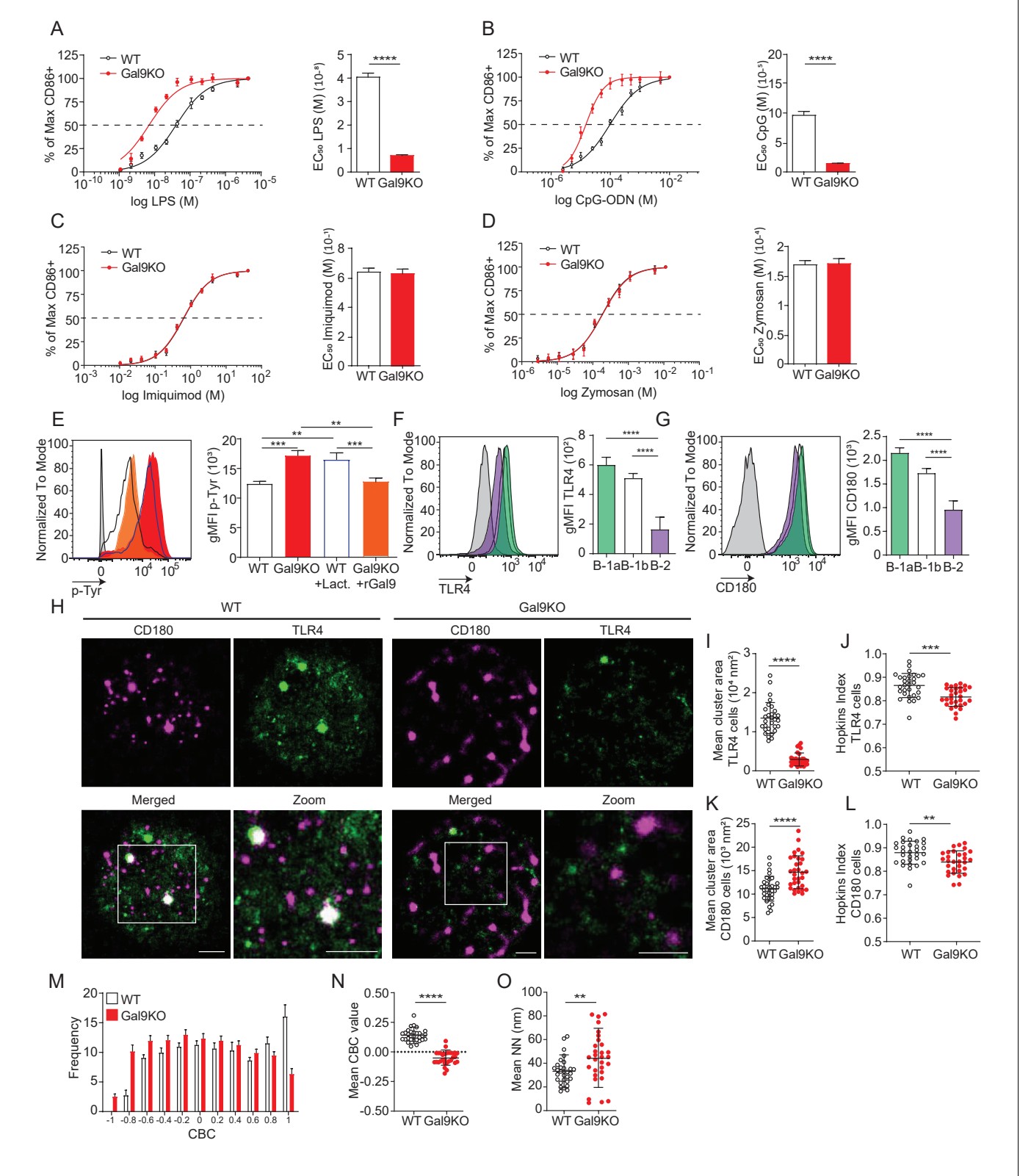

**Figure 6.** Gal9 regulates LPS-mediated activation by restraining signal transduction through interactions with TLR4 and the negative co-receptor CD180. (**A**) Proportion of CD86 expressing cells of B-1a cells from WT (black) and Gal9KO (red) mice stimulated with titrated concentrations of LPS (left), $EC_{50}$ (right). (**B**) Titration with CpG-ODN as in **A** (left), $EC_{50}$ (right). (**C**) Titration with Imiquimod as in (**A**) (left), $EC_{50}$ (right). (**D**) Titration with Zymosan as in (**A**) (left), $EC_{50}$ (right). (**E**) Representative histograms of p-Tyr levels at 10 min in B-1a cells from WT (black, open) or Gal9KO (red, filled) mice, or WT

*Figure 6 continued on next page*

Figure 6 continued

B-1a cells treated with lactose (blue, open) or Gal9KO B-1a cells treated with rGal9 (orange, filled) and stimulated with LPS for 10 min (left) summary gMFI of p-Tyr expression (right). (F) Representative histograms (left) and summary gMFI (right) of TLR4 detection on rGal9-coated beads incubated with lysates from different B cell subsets, as indicated. (G) Representative histogram (left) and summary gMFI (right) of CD180 detection as in (F). (H) Reconstructed dual-dSTORM images of CD180 (magenta) and TLR4 (green) on the surface of B-1a cells from WT and Gal9KO mice, ROI (3 µm × 3 µm) used for analysis is expanded in zoom. Scale bars represent 1 µm. (I) Quantification of TLR4 cluster area from dSTORM images. (J) Quantification of TLR4 clustering tendency using Hopkins Index. (K) Quantification of CD180 cluster area from dSTORM images. (L) Quantification of CD180 clustering tendency using Hopkins index. (M) Frequency distribution of coordinate-based colocalization (CBC) between TLR4 and CD180. (N) Mean CBC value per ROI. (O) Mean distance between nearest neighbors. Data in A–G are representative of nine biological replicates over three independent experiments. Data in H and I are representative of 30 ROIs acquired over at least three independent experiments. Data represent mean ± SEM. Statistical significance for A, D, I, J, K, and L was assessed by Mann–Whitney, statistical significance for E, F, and G was assessed by Kruskal–Wallis **p≤0.01, *** p≤0.001, ****p<0.0001.

The online version of this article includes the following figure supplement(s) for figure 6:

**Figure supplement 1.** Gal9 does not alter LPS responses in B-1b cells.

mediate this phenotype, we performed pull down assays with rGal9-coated beads incubated with B-1a or B-1b cell lysates, as previously. We find that Gal9 binds TLR4 and the inhibitory molecule CD180 on B-1 cells, and to a lesser extent on B-2 B cells (*Figure 6F,G*), consistent with previous mass spectrometry analysis of Gal9 ligands in B2 B cells (*Cao et al., 2018*).

To assess the impact of Gal9 on TLR4 nano-scale organization, we again performed dual-color dSTORM to identify single molecule localizations of TLR4 and CD180 on the surface of B-1 cells from WT and Gal9KO mice (*Figure 6H*). In the absence of Gal9 the mean cluster area and clustering tendency of TLR4 is decreased on B-1a cells (*Figure 6I–J*), but interestingly, Gal9 has no affect on the organization of TLR4 in B-1b cells (*Figure 6—figure supplement 1F-H*). Similarly, the clustering tendency of CD180 is decreased in the absence of Gal9 in B-1a cells; however, the area of CD180 clusters is larger (*Figure 6H,K–L*), whereas deficiency of Gal9 has no effect on the organization of CD180 in B-1b cells (*Figure 6—figure supplement 1F,I–J*). To assess whether Gal9 mediates close association of TLR4 with the inhibitory co-receptor CD180, we performed CBC analysis as previously described. The mean CBC value of TLR4 and CD180 is decreased, and mean nearest-neighbor increased, in B-1a cells from Gal9KO mice compared to WT mice (*Figure 6M–O*), however there is no difference in B-1b cells (*Figure 6—figure supplement 1K,M*). Taken together, these data demonstrate that Gal9 increases the association of TLR4 and CD180, specifically on the surface of B-1a cells, but not B-1b cells, to modulate TLR4 signaling.

## B-1a cells facilitate autoantigen delivery to splenic B cells

TLR stimulation of B-1 cells leads to rapid ASC differentiation (*Ha et al., 2006*). So, we asked if Gal9-mediated regulation of TLR signaling influences ASC development. To address this, we isolated PerC B-1 cells from WT and Gal9KO mice and sorted B-1a cells. We stimulated sorted cells ex vivo with 0.5 µg/mL LPS and 0.5 µg/mL anti-IgM and measured ASC differentiation by CD138 expression 3 days post-stimulation. We find that B-1a cells from Gal9KO mice more readily adopt an ASC phenotype following stimulation, which results in more secreted antibody in the culture supernatant (*Figure 7A–D*).

As B-1a-derived autoantibodies are thought to aid in clearance of cellular debris (*Duan and Morel, 2006*), we therefore asked if enhanced sensitivity of B-1a cells to LPS could have an impact in vivo. To address this, we injected mice with a low (0.05 µg/mL) or high dose (1 µg/mL) of LPS by intraperitoneal (IP) injection and assessed B cell activation 24 hr post-injection. Following low-dose LPS injection, upregulation of CD86 is enhanced in Gal9-deficient B-1a cells, but not B-1b cells (*Figure 7—figure supplement 1A*). We then asked if these activated B-1a cells are secreting autoantibodies into circulation. Indeed, we find a greater fold increase in both PC-specific IgM and IgG in Gal9KO mice 24 hr post low-dose LPS injection (*Figure 7—figure supplement 1B*).

We then asked if B-1a cell-derived immune complexes that form following LPS injection carry autoantigens into secondary lymphoid organs. To address this, we IP injected mice with low-dose LPS followed by CFSE-labeled apoptotic bodies (ApoBs) by intravenous injection 24 hr later. At 12 hr post-transfer, we looked for ApoB capture by subcapsular sinus (SCS) macrophages in the spleen of WT or Gal9KO mice. We find that ICs formed following LPS injection are capable of capturing

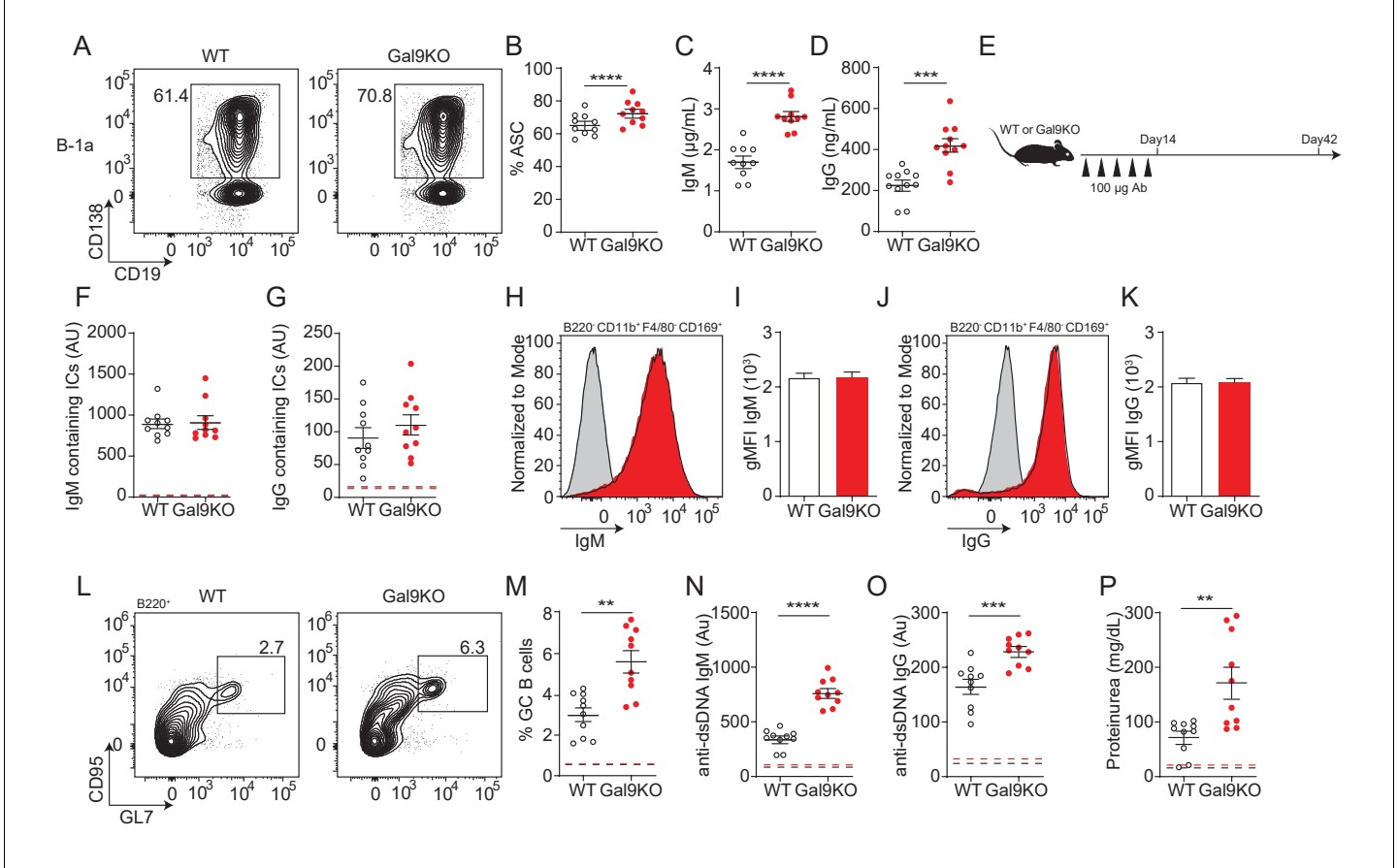

**Figure 7.** Transfer of B-1a derived antibodies leads to autoimmunity in mice. (A) Representative plots of ASC development in sorted WT and Gal9KO B-1a cells cultured for 3 days in presence of 0.5 µg/mL LPS and 0.5 µg/mL IgM. (B) Summary proportion of ASC differentiation in sorted WT (black, open) and Gal9KO (red, filled), as in (A). (C) Secreted IgM titers, and (D) Secreted IgG titers in the culture supernatant, as in (A). (E) Schematic of in vivo B-1-derived antibody transfer experiment. (F) Titers of IgM and (G) IgG-containing ICs in circulation at day 14, mean baseline titers shown as dashed line. (H) Representative histograms and (I) summary gMFI of IgM expression on splenic subcapsular sinus macrophages at day 14. (J) Representative histograms and (K) summary gMFI of IgG expression on splenic subcapsular sinus macrophages at day 14. (L) Representative plots and (M) summary proportion of GC B cell differentiation in the spleen at day 42, mean baseline frequency shown as dashed lines. (N) Titers of dsDNA-specific IgM and (O) IgG in sera at day 42, mean baseline titers shown as dashed line. (P) Protein secretion in the urine of mice at day 42, mean baseline level shown as dashed line. Data are representative of 10 biological replicates over two independent experiments. Statistical significance was assessed by Mann–Whitney, *p≤0.05, **p≤0.01, ***p≤0.001, ****p<0.0001.

The online version of this article includes the following figure supplement(s) for figure 7:

**Figure supplement 1.** Gal9KO B-1a cells have enhanced activation to low LPS stimulation.

CFSE-labeled ApoB and recruiting them to SCS macrophages in the spleen (*Figure 7—figure supplement 1C–E*). To determine whether autoantigen delivery to SCS macrophages was directly driven by increased antibody secretion by B-1a cells, we isolated secreted antibodies from sorted B-1a and B-1b cultures stimulated ex vivo and injected these antibodies into mice every 3 days over a 2-week period (*Figure 7E*). Following 2 weeks of antibody transfer, we detect ICs in circulation as well as in the spleens of mice receiving B-1a-derived antibodies in both WT and Gal9KO recipient mice (*Figure 7F–K*).

To assess whether enhanced delivery of ICs bearing autoantigens facilitates autoimmunity, we measured splenic GC B cells 4 weeks following final antibody transfer. Following transfer of B-1a-derived antibodies, GCs formed in the spleen and autoantibodies are detected in circulation (*Figure 7L–O*). Additionally, we detect low levels of protein secretion into the urine at this time point, suggesting that these mice begin to develop nephritis (*Figure 7P*). These observations are further exacerbated in Gal9KO mice, which have a reduced threshold for B cell activation, and

subsequently develop a more exaggerated autoimmune response (*Figure 7L–P*). Collectively, these data demonstrate that Gal9 regulates BCR-driven and unique TLR-driven B-1a responses by regulating the association of these receptors with negative co-receptors, loss of which enhances B-1a-derived antibody production and consequently autoantigen delivery to splenic B cells to drive autoimmunity.

## Gal9 regulates disease onset in the NZB/W model of murine lupus

Loss of Gal9 leads to enhanced TLR4 signal transduction in B-1a cells, allowing for enhanced activation and systemic antibody secretion. Similarly, loss of Gal9 decreases the threshold of B cell activation, allowing for B cell responses against low-affinity antigens. We therefore asked if loss of Gal9 would lead to more rapid disease onset in the NZB/W F1 model system for lupus-like disease in mice. To address this, we backcrossed the Gal9KO allele onto the NZB and NZW backgrounds for five generations. We subsequently crossed heterozygous NZB and NZW mice to generate WT and Gal9-deficient NZB/W F1 littermate controls, which develop spontaneous autoimmunity with age. Indeed, we find that Gal9-deficient mice produce anti-dsDNA IgG antibodies earlier in life compared to WT littermate controls (*Figure 8A*). Furthermore, mice lacking Gal9 expression develop accelerated proteinuria compared to WT controls (*Figure 8B*). Consistent with these observations, Gal9-deficient mice have a larger population of splenic GC B cells early in disease progression compared to WT littermates (*Figure 8C,D*). We see a similar increase in the proportion of splenic ASCs in Gal9-deficient mice early in disease onset (*Figure 8E,F*). These ASCs skew toward B220⁻ long-lived ASCs, and consistent with this, we see enhanced immune complexes in circulation in mice lacking Gal9 (*Figure 8G,H*). We then asked if loss of Gal9 would lead to an expanded pool of B-1a cells in the peritoneal cavity, as we observed in the C57BL/6 background. Indeed, NZB/W F1 mice deficient in Gal9 have a larger population of peritoneal B-1a cells (*Figure 8I*). If these expanded B-1a cells are contributing to disease progression, we would expect to see signs of their activation such as migration to secondary lymphoid tissues. Indeed, there is an increased proportion of B-1 cells in the spleen in Gal9-deficient mice (*Figure 8J*), and this population is enriched for B-1a cells (*Figure 8K*). Furthermore, loss of Gal9 leads to enhanced production of B-1a-derived PC-specific IgM antibodies detectable in the serum (*Figure 8L*). Together, these data demonstrate that Gal9 regulates the onset and progression of lupus-like disease in mice. Loss of Gal9 leads to accelerated B cell-driven autoimmunity, marked by GC B cell and ASC formation as well as B-1a cell expansion, migration, and antibody secretion.

## Discussion

Our data indicate that Gal9 regulates IgM signal transduction and thereby sets a threshold of agonist required for productive activation. Our previous work demonstrated that Gal9 binds exclusively to IgM-BCR, with no evidence to support the interaction between Gal9 and other BCR isotypes. Through binding to the IgM-BCR and important co-receptors of BCR signaling, Gal9 alters the nano-scale organization of the plasma membrane (*Cao et al., 2018*). Herein, we show that loss of this regulation leads to spontaneous autoimmunity in mice, consistent with an expanded population of B-1a cells. Interestingly, we identify Gal9 as a regulator of B-1a cell activation through interactions with IgM-BCR, CD5, TLR4, and CD180, likely through interactions between the carbohydrate recognition domain of Gal9 with specific glycan-moieties on these glycoproteins. However, the specific glycosylation sites and structural constraints of these interactions remain to be fully resolved. Branching of N-glycans has been shown to regulate aspects of B cell activation with respect to TLR-induced B cell functional responses (*Mortales et al., 2020*). Branched complex N-glycans serve as a ligand for galectins, including Gal9. Our findings, in conjunction with the growing appreciation for the role of glycan modifications in regulating B cell responses, highlight the complex nature of B cell activation (*Mortales et al., 2020*). We find that loss of Gal9 leads to decreased colocalization between receptors IgM/TLR4 and their cognate regulators, CD5 and CD180, on the surface of B-1a cells. This coincides with enhanced signal transduction and a reduced threshold for activation. We demonstrate that B-1a-derived antibodies promote development of autoimmunity in mice, likely due to their capacity to bind and transport autoantigens to secondary lymphoid tissues.

Our findings suggest that Gal9 modulates antigen affinity discrimination in B cells, a critical regulatory step in peripheral tolerance (*Basten and Silveira, 2010*). Unchecked signal transduction leads

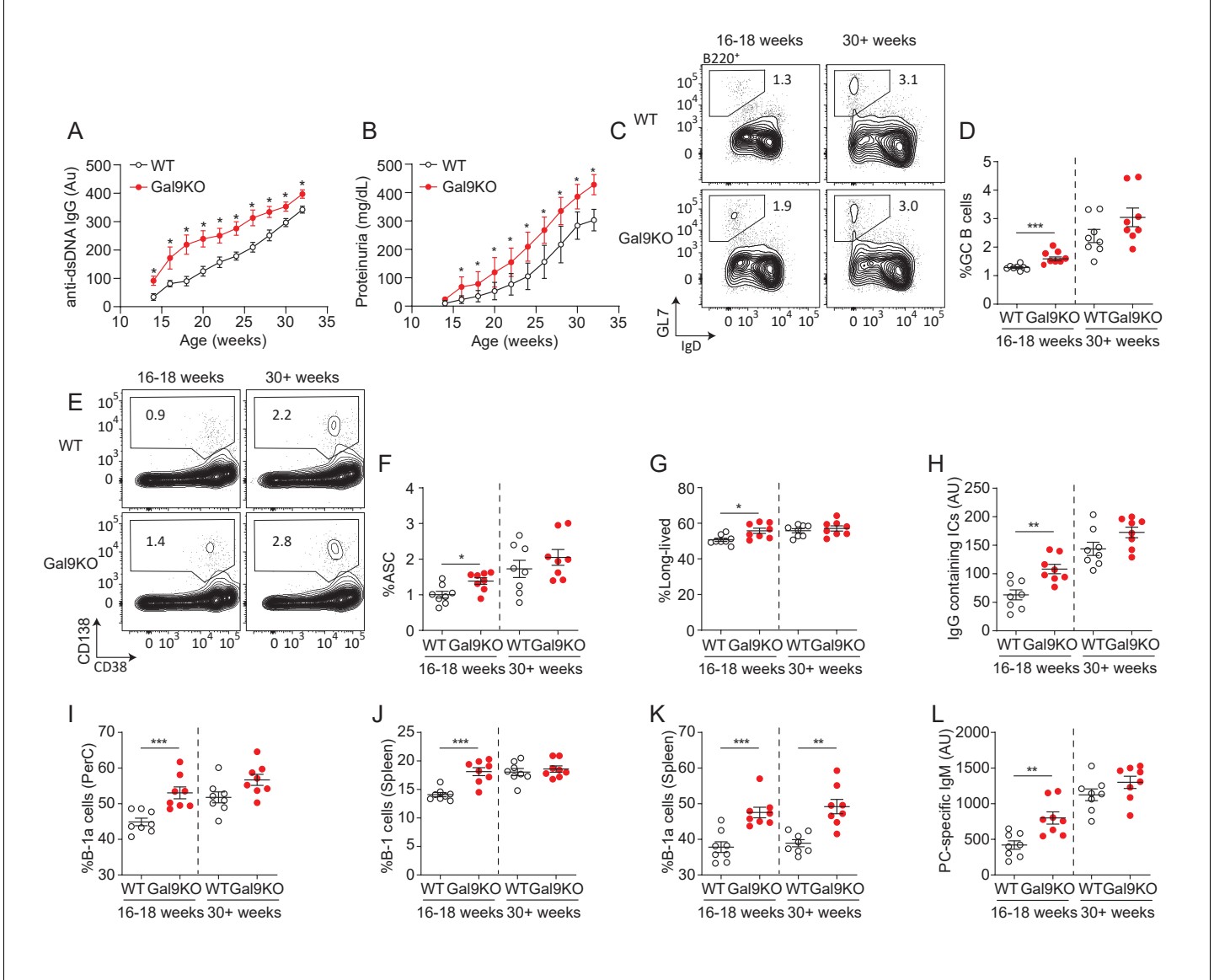

**Figure 8.** Loss of Gal9 in NZB/W mice leads to accelerated autoimmunity. (**A**) Titers of dsDNA-specific IgG in sera of NZB/W mice (as indicated) monitored over time. (**B**) Protein secretion in the urine of NZB/W mice monitored over time. (**C**) Representative plots and (**D**) summary proportion of splenic GC B cells in NZB/W mice (as indicated). (**E**) Representative plots and (**F**) summary proportion of antibody secreting cells (ASC) in the spleen of NZB/W mice. (**G**) Proportion of long-lived ASC lacking B220 expression. (**H**) Circulating IgG-containing immune complex (ICs) titers. (**I**) Proportion of CD5+ B-1a cells in the peritoneal cavity of NZB/W mice. (**J**) Proportion of CD19+ CD43+ B-1 cells in the spleen of NZB/W mice. (**K**) Proportion of splenic CD5+ B-1a cells. (**L**) Circulating phosphoryl-choline specific IgM titers in NZB/W mice. Data in (**A**) and (**B**) are representative of at least 14 biological replicates, data in (**C–L**) are representative of eight biological replicates. Data represent mean and SEM. Statistical significance was assessed by Mann–Whitney, *p≤0.05, **p≤0.01, ***p≤0.001.

to autoimmunity in mice (*Zikherman and Lowell, 2017*). In fact, aberrations in CD19 or BTK signaling, both of which are known to play important roles in defining this threshold of activation, lead to autoantibody production (*Zikherman and Lowell, 2017*). Additionally, it is well established that negative regulators such as CD45 and CD22 play central roles in restraining B cell activation and are critical for antigen affinity discrimination in vivo (*Zikherman and Lowell, 2017*). We have previously shown that Gal9 binds directly to IgM-BCR and the negative regulators CD45 and CD22, thereby dampening BCR signal transduction. Here, we demonstrate that by fine-tuning these signal transduction networks, Gal9 coordinates affinity discrimination of B cells and that loss of this regulation leads to the development of autoimmunity in mice.

Our data indicates that Gal9 suppresses B cell responses to low-affinity autoantigens; yet the role of Gal9 in autoimmunity may be cell type specific and multifaceted. For example, administration of rGal9 to NZB/W F1 mice prior to disease manifestation led to a reduction in disease severity (*Panda et al., 2018*). In this work, the authors elucidated a role for Gal9 in restraining TLR7 and TLR9 responses in pDCs, important drivers of type I IFNs and the inflammatory profile that supports autoimmunity (*Panda et al., 2018*). Consistent with this, we show that loss of Gal9 leads to accelerated spontaneous autoimmunity in NZB/W F1 mice. Mechanistically, we show that Gal9 regulates TLR4 and TLR9 responses specifically in B-1a cells, but not B-1b cells, although unlike in pDCs, we see no effect of Gal9 on TLR7 responses in B-1 cells. This cell type disparity in the requirement or role of Gal9 in regulating receptor signaling and therefore cellular activation may be due to differential protein glycosylation and expression of Gal9 between different cell types and subsets. For example, Gal9 seems to be essential for development of autoimmune disease in the pristane-induced model, where Gal9KO mice have significantly reduced disease severity (*Moritoki et al., 2013*). This may result, in part, from impaired DC activation, which is essential for disease onset in pristane-induced autoimmunity (*Richard and Gilkeson, 2018*). In this case, intracellular Gal9 interacts with the cortical actin cytoskeleton to facilitate phagocytosis and DC activation (*Querol Cano et al., 2019*), and therefore, genetic deletion of Gal9 may be detrimental for DC-dependent models of autoimmunity. The growing compendium of data suggest that Gal9 may have pleotropic effects that result in context-dependent and cell-type-specific regulation (*Panda and Das, 2018*).

B-1a cells are expanded in several models of autoimmunity; however; their role in disease pathology remains controversial (*Duan and Morel, 2006*). Their autoreactive repertoire and heightened capacity for T cell activation suggest that these cells may be drivers of autoimmunity. Here we demonstrate that Gal9 regulates activation of B-1a cells specifically through regulating IgM and TLR4 signal transduction by increasing the colocalization of these receptors with inhibitory co-receptors CD5 and CD180, respectively. Furthermore, we show that Gal9 restrains ASC differentiation of B-1a cells and antibody production. Transfer of apoptotic bodies into mice is sufficient to drive autoantibody production, likely through increasing the density of autoantigens in secondary lymphoid organs (*Mevorach et al., 1998*). Similarly, mutations that impair efferocytosis of circulating immune complexes can lead to autoimmunity (*Abdolmaleki et al., 2018*). Here, we demonstrate that enhanced production of B-1a-derived antibodies increases circulating immune complexes in mice and that these immune complexes transit to secondary lymphoid organs and are captured by SCS macrophages. This then results in the development of autoimmune disease, highlighting the potential pathogenic role of B-1a cells in the onset of autoimmunity. Notably, the development of autoimmune disease is exacerbated in Gal9KO mice, in large part due to the reduced threshold of activation of both B-1a and B-2 B cells.

Our findings suggest that changes in the level of Gal9 could be sufficient to reduce the threshold of IgM-BCR and distinct TLR signaling, driving strong BCR and TLR signaling upon binding of low-affinity and/or low-density ligands. There is ample evidence that expression of Gal9 is altered in a variety of disease conditions, including viral infection, inflammation, and cancer (*Chagan-Yasutan et al., 2013*; *Chou et al., 2018*). In the context of autoimmunity, emerging evidence has suggested that Gal9 is elevated in the serum (*Matsuoka et al., 2020*); however, these results should be interpreted with caution, as degraded Gal9 in the serum could obscure the interpretation of these findings (*Robinson et al., 2019*). Thus, it is not clear whether elevated serum Gal9 is reflective of functional Gal9 or whether it represents a decrease in bound and functional Gal9. Moreover, glycosylation of cell surface proteins is regulated by a variety of factors including cell metabolism, ER and Golgi nucleotide sugar-donor transporters, and the expression and activity of glycosyltransferases, glycosidases, and glycan-modifying enzymes. Indeed, selective glycosyltransferase expression may be a mechanism to disparately regulate that activity of galectin binding in B cells, as recent work has demonstrated upregulation of the I-branching enzyme GCNT2 in GC B cells, but not naïve and memory B cells, with a concomitant decrease in Gal9 binding (*Giovannone et al., 2018*). How these varied factors may be altered in the context of autoimmunity to modify the glycan profile of BCR, TLR, and key regulatory co-receptors, and thus galectin binding, has yet to be elucidated.

In summary, the present work demonstrates that Gal9 plays a critical role in tuning the outcome of receptor–ligand interactions. Loss of Gal9 leads to robust activation of B cells to low-affinity and low-density antigens. Additionally, Gal9 regulates BCR and unique TLR-driven B-1a cell activation through a unique molecular mechanism, important in the regulation of autoantigen transit to

secondary lymphoid organs. Collectively, these data demonstrate that Gal9 may serve as a therapeutic target to help mitigate the onset of autoimmunity.

## Materials and methods

### Mice

C57BL/6 (WT) mice were obtained from Charles River, *Lgals9*$^{-/-}$ (Gal9KO) mice were obtained from Steven Beverley (Washington University) on behalf of The Scripps Research Institute, *Nr4a1*-eGFP (Nur77-eGFP; C57BL/6-Tg(Nr4a1-EGFP/cre)820Khog/J; Stock No. 016617), NZB/BlNJ (Stock No. 000684), NZW/LacJ (Stock No. 001058), MD4 (C57BL/6-Tg(IghelMD4)4Ccg/J; Stock No. 002595), and µMT (B6.129S2-*Ighm*$^{tm1Cgn}$/J; Stock No. 002288) mice were obtained from The Jackson Laboratory. MD4/ML5 (B6.Ig/sHEL) mice were obtained from Joan Wither (University of Toronto, Krembil Research Institute, University Health Network). Mice were used at 2–6 months of age for all functional and biochemical experiments, and mice were age- and sex-matched within experiments. Mice were used at 8–14 months of age for characterization of spontaneous autoimmunity. Mice were housed in specific pathogen-free animal facility at University of Toronto Scarborough, Toronto, Canada. All procedures were approved by the Local Animal Care Committee at the University of Toronto Scarborough, Animal Use Protocols 20012282 and 20011481.

### Cell lines

HEp-2 cells, verified to be contaminated with HeLa cells, were obtained from ATCC and were certified mycoplasma-negative and used acutely. OP9-R7FS cells were kindly provided by Juan Carlos Zúñiga-Pflücker (University of Toronto, Sunnybrook Research Institute) and authenticated by mouse STR profiling by ATCC and mycoplasma tested via MycoAlert Mycoplasma Detection Kit (Lonza). Cells were maintained at 37°C with 5% $CO_2$ in DMEM containing 10% heat-inactivated fetal bovine serum (FBS, Wisent), 100 U/mL penicillin and streptomycin (Gibco), and 50 µM 2-mercaptoethanol (Amresco).

### Lysozyme isolation

Lysozymes were isolated from egg whites by ion exchange. Egg whites were diluted in PBS and filtered through 30 kDa filter to restrict protein size (Amicon). Fraction <30 kDa was added to 100 mM ammonium acetate buffer (pH 9.0) and incubated with carboxymethylcellulose (CMC, Sigma-Aldrich). Unbound proteins were washed with ammonium acetate buffer, and lysozymes were then eluted with 400 mM ammonium carbonate buffer (pH 9.2). Isolated lysozymes were diluted in PBS and concentrated using a centrifugation column (Amicon). Purified lysozymes were then mono-biotinylated (*Fleire and Batista, 2009*) using EZ-Link NHS-LC-LC-Biotin (Thermo Fisher Scientific) according to the manufacturer's protocol. Mono-biotinylation was confirmed by flow cytometry as described (*Fleire and Batista, 2009*).

### Cell isolation

Splenocytes were isolated from mice using a 70 µm cell strainer and phosphate-buffered saline (PBS, pH 7.4, Wisent) to form a single-cell suspension. Bone marrow was isolated by centrifugation of dissected long bones. Peritoneal cells were isolated by peritoneal lavage. B cells were purified using negative magnetic isolation kit (Miltenyi Biotec Inc or Stem Cell Technologies) according to the manufacturer's protocol.

### B cell culture and stimulation

Primary B cells were cultured in RPMI1640 (Gibco) containing 10% heat-inactivated fetal bovine serum (FBS, Wisent), 100 U/mL penicillin and streptomycin (Gibco), and 50 µM 2-mercaptoethanol (Amresco) at 37°C with 5% $CO_2$. For activation experiments: cells were stimulated with titrated concentrations of agonists (as indicated) for 16 hr, then stained with fluorescently conjugated antibodies targeting CD86, CD69, and MHCII (BioLegend), and analyzed by flow cytometry. For BCR internalization experiments: primary B cells were stimulated with agonists (as indicated) in RPMI1640 at 37°C. Cells were then fixed at indicated time points in 2% paraformaldehyde stained with fluorescently conjugated antibody Fab fragments targeting IgM (Jackson Immunoresearch) and analyzed by flow

cytometry. For signaling experiments: primary B cells were stimulated with agonists (as indicated) in RPMI1640 at 37°C. Cells were fixed at indicated time points in 2% paraformaldehyde and permeabilized on ice in 100% methanol at −20°C. Cells were washed 3× in PBS and then stained for total tyrosine phosphorylation (clone 4G10, Sunnybrook), and cells were subsequently stained with secondary antibodies (Invitrogen) as well as phenotyping markers (BioLegend). Signaling was measured by flow cytometry. To detect internalization of lysozyme, primary MD4 B cells were stimulated with mono-biotinylated lysozyme in RPMI 1640 at 37°C. Cells were fixed in 2% paraformaldehyde, stained with PE-conjugated streptavidin (BioLegend) to occupy mono-biotinylated lysozyme on the cell surface. Cells were then permeabilized in 0.1% Saponin permeabilization buffer (0.1% saponin, 3% FBS, 0.1% sodium azide, in PBS) and stained with Alexa Fluor-633-conjugated streptavidin (Invitrogen) to detect intracellular lysozyme by flow cytometry. For experiments using OP9 stromal cells: OP9 cells were labeled with 5 µM CFSE (Invitrogen). Immune complexes containing mono-biotinylated lysozyme and anti-sera generated in mice were formed in vitro, then incubated with OP9 stromal cells in complete media in suspension, then washed with PBS, and stained with Alexa Fluor-633-conjugated streptavidin (Invitrogen). Amount of lysozyme was determined by flow cytometry using fluorescence quantitation beads according to the manufacturer's protocol (Bangs Laboratories Inc). Density of molecules was then determined using the average surface area of a cell determined by confocal imaging. OP9 cells were then seeded into wells and centrifuged to draw cells to the bottom. Primary MD4 B cells were added to the top of the wells and allowed to settle onto OP9 cells. Cells were fixed using 2% paraformaldehyde and permeabilized on ice in 100% methanol at −20°C. Cells were washed 3× in PBS and then stained for total tyrosine phosphorylation (clone 4G10, Sunnybrook), followed by secondary antibodies (Invitrogen) as well as anti-B220-Pacific Blue (BioLegend).

## Planar lipid bilayer

Artificial planar lipid bilayers were prepared by spreading liposomes in FCS2 chambers (Bioptechs). 1,2-Dioleoyl-sn-glycero-3-phosphocholine liposomes (Avanti Polar Lipids, Inc) were mixed with 2.5 × $10^{-4}$% biotinylated liposomes (N-Biotinylated Cap PE: Avanti Polar Lipids, Inc). Alexa Fluor 633-streptavidin (1:1000 v/v; Life technologies) was used to tether monobiotinylated lysozymes. Cells were injected into FCS chambers in chamber buffer (0.5% FBS, 2 mM MgCl$_2$, 0.5 mM CaCl$_2$, and 1 g/L D-glucose in PBS). Freshly isolated splenocytes from WT and Gal9KO MD4 mice were allowed to spread and interact with the bilayer for 90 s and then fixed with 2% PFA for 15 min at 37°C. Cells were imaged by TIRF microscopy and analyzed using ImageJ. Total contact area and total fluorescence intensities were calculated for each cell. Fluorescence intensity was converted into molecular count using a fluorescence standard (*Fleire and Batista, 2009*) (Quantitation set, Bangs Laboratories).

## Autoantibody detection

HEp-2 cells were grown on glass bottom dishes and fixed with 2% paraformaldehyde. Cells were then stained with sera isolated from mice (1:50) at 4°C. Cells were then washed, and autoantibody binding was detected using secondary anti-mouse IgM or IgG (Jackson ImmunoResearch). Cells were imaged by spinning disc confocal microscopy (Quorum Technologies) consisting of an inverted fluorescence microscope (DM16000B; Leica) equipped with a 63×/1.4 NA oil-immersion objective and an electron-multiplying charge-coupled device (EMCCD) camera (ImagEM; Hamamatsu). Z-section images were collapsed into a Z-projection, and mean fluorescence intensity (MFI) was determined by dividing the total intensity of the field of view by the number of nuclei using ImageJ.

## Antigen-specific ELISAs

Antibodies were quantified using IgM and IgG quantitation sets (Bethyl Laboratories Inc) according to the manufacturer's protocol with some adjustments: Thermo Scientific Nunc MaxiSorp 96-well ELISA plates were coated with 50 µg/mL calf thymus DNA (Thermo Fisher Scientific), phosphorylcholine (Sigma-Aldrich), hen egg lysozyme (Sigma Aldrich), glomerulus extract isolated from primary mouse glomeruli lysed in NP40 lysis buffer, or anti-C3 (clone K13/16, BioLegend for immune complex determination) in sodium carbonate coating buffer. Sera was diluted at 1:50 for quantification. Signal was quantified against IgG/IgM standard and reported as absorbance units (AU).

## Lactose/rGal9 treatment

Primary murine B cells were incubated in 20 mM β-lactose or 0.1 μM recombinant Gal9 (R&D Systems) in RPMI1640 at 37°C for 30 min. Cells were washed with PBS and resuspended in RPMI1640. Cells were then stimulated as described above.

## Pull down assay

Isolated B cells from Gal9KO mice were lysed in NP40 lysis buffer (50 mM HEPES pH 8.0, 150 mM NaCl, 5 mM dithiothreitol, 5 mM EDTA, 0.1% Nonidet-P40, Roche cOmplete mini EDTA-free protease inhibitor, and 1 mM phenylmethylsulfonylfluoride). Lysates were cleared by centrifugation at 21,000 × g for 10 min at 4°C. Cell lysates were incubated with anti-FLAG beads (Sigma-Aldrich) conjugated with 10 μg FLAG-tagged rGal9, kindly provided by the Ostrowski Lab (University of Toronto) for 3 hr at 4°C. Beads were washed with lysis buffer and blocked in 5% horse serum in PBS. Bound proteins were detected by staining with fluorescently conjugated antibodies (BioLegend) and measured by flow cytometry. Anti-FLAG beads alone (without rGal9) were incubated with lysates and served as a negative control of background adsorption.

## dSTORM

### Sample preparation

Primary murine peritoneal cells were washed in PBS and blocked in 1:50 FC block (clone S17011E) on ice for 20 min. Cells were then stained with 2 μg/mL of Alexa Fluor 647 labeled anti-CD5 antibody or anti-CD180 antibody (BioLegend) and 2 μg/mL of Alexa Fluor 488 labeled anti-IgM Fab fragment (Jackson ImmunoResearch) or anti-TLR4 (Invitrogen) on ice for 30 min. Cells were washed 2× with PBS and resuspended in PBS. Cells were incubated at 37°C for 5 min and then injected into warmed FCS2 chambers coated with 2 μg/mL anti-MHCII antibody (clone M5/114; Sunnybrook Research Institute). Cells were spread at 37°C for 10 min and then fixed with 4% paraformaldehyde with 0.2% glutaraldehyde in PBS for 45 min at room temperature.

### Image acquisition and reconstruction

Chambers wereincubated with dSTORM imaging buffer: 0.1 M β-mercaptoethylamine (MEA, Sigma-Aldrich), 3% (v/v) OxyFluorTM (Oxyrase Inc), 20% (v/v) sodium DL-lactase solution (Sigma-Aldrich) adjusted to pH 8.3, containing Fiducial markers (100 nm Tetraspeck Fluorescent Microspheres, Invitrogen). dSTORM imaging was performed on a TIRF microscope (Quorum Technologies) based on an inverted microscope (DMI6000C, Leica), HCX PL APO 100X/1.47 oil immersion objective and Evolve Delta EMCCD camera (Photometrics). Alexa Fluor 647 was imaged first, photoconversion was achieved with 633 nm laser (intensity ranged from 60 to 80 mW/cm$^2$) illumination and simultaneous illumination with 405 nm laser (intensity range from 5 to 40 mW/cm$^2$) increased the rate of conversion from the dark state. Subsequently, Alexa Fluor 488 was imaged, photoconversion was achieved with the 488 nm laser (intensity ranged from 60 to 80 mW/cm$^2$). Eight thousand images were acquired with 30 ms exposure and a frame rate of 33 frames/s. Image reconstruction was performed using the ThunderSTORM plugin for ImageJ, using the parameters reported previously (*Cao et al., 2018*). Camera setup was pixel size 101.5 nm, photoelectron per A/D count 2.4, base level [A/D count] 414, EM gain of 200. Image filtering was applied using a wavelet filter (B-Spline) with a B-Spline order of 3 and B-Spline scale of 2.0. Approximate localization of molecules was detected by local maximum method with a peak intensity threshold of std (Wave.F1) and a connectivity of 8-neighborhood. Sub-pixel localization of molecules was identified by fitting the point spread function to an integrated Gaussian distribution using weighted least squares method with a fitting radius of 3 pixels. Multiple emitter fitting analysis was used to estimate the number of molecules detected as a single blob with a maximum of five molecules per fitting region. Molecules were fitting assuming the same intensity. Super-resolution images were rendered with a pixel size of 20 nm.

### Processing and analysis

Reconstructed images were processed using the built-in method in the Thunderstorm plugin. Duplicate localizations of a single molecule in a given frame were removed based on uncertainty radium of localization. Localizations with an uncertainty > 20 nm were filtered out. Isolated localizations with fewer than two neighbors in a 50 nm radius were removed. The images were drift corrected using

fiducial markers, and molecules that appear within 20 nm in multiple frames were merged. Fiducial markers were used to align the two channels post processing using the ImageJ plugin TurboReg, as necessary. For each cell, a 3 × 3 µm region was selected in the middle of the cell for analysis. Cluster area was calculated using the density-based spatial clustering of applications with noise (DBSCAN) algorithm in the SMoLR package in R. The Hopkins index was calculated using the Spatstat package in R. Colocalization was assessed using the coordinate-based colocalization (CBC) analysis tool in ThunderSTORM. Each localization is assigned an individual colocalization value based on individual distribution functions of that species and weighted by the distance of nearest neighbors in the local environment. For IgM and CD5 CBC analysis, a search radius of 60 nm was used, and for TLR4 and CD180 CBC analysis, a search radius of 50 nm was used, based on the radius of IgM and TLR4 nano-clusters, respectively. CBC values then reflect the degree of colocalization between two channels on a scale between +1 (perfectly colocalized) to −1 (perfectly excluded).

## Ex vivo antibody production from B-1 cells

Primary murine peritoneal cells were sorted into B-1a and B-1b cells using MACS B-1a selection kit (Miltenyi Biotec Inc). Sorted cells were cultured in RPMI1640 containing 10% heat-inactivated fetal bovine serum (FBS, Wisent), 100 U/mL penicillin and streptomycin (Gibco), and 50 µM 2-mercaptoe-thanol (Amresco) for 3 days with 0.5 µg/mL LPS (Invivogen) and 0.5 µg/mL anti-IgM F(ab')$_2$ (Jackson ImmunoResearch). Cultures were collected, and cells were pelleted by centrifugation. Culture super-natants containing secreted antibodies were collected, and antibodies were purified by filtration using a 100 kDa centrifugation filter (retaining the > 100 kDa fraction). Antibody composition was determined by ELISA using mouse IgM/IgG quantitation set (Bethyl Laboratories), according to the manufacturer's protocol. Total concentration of antibody was determined by BCA assay (Thermo Fisher Scientific).

## Transfer of B-1a-derived antibodies

Mice were transferred 100 µg of total antibody by intraperitoneal injection every 3 days over a 2-week period. At day, 14 mice were euthanized and a single cell suspension was prepared from the spleen using enzymatic digestion (DNAse/Collagenase). Cells were stained with fluorescently labeled antibodies and measured by flow cytometry. Additionally, mice were assessed 4 weeks following the last transfer for signs of autoimmunity as described above.

## Generation of apoptotic bodies

Primary mouse thymocytes were isolated and formed into a single-cell suspension using a 70 µm cell strainer. Cells were labeled with 10 µm of CFSE and washed as described above. Cells were then cul-tured for 2 days in RPMI1640 containing 100 U/mL penicillin and streptomycin (Gibco) and 50 µM 2-mercaptoethanol (Amresco) and 5 µg/mL puromycin (Thermo Fisher Scientific) to induce apoptosis. Cultures were collected and washed with PBS; intact cells were pelleted by gentle centrifugation at 300× g for 5 min. Cell supernatants containing ApoB were transferred into mice by intravenous injection.

## LPS activation in vivo

Mice were injected with 1 µg/mL or 0.05 µg/mL of LPS by intraperitoneal injection to activate B-1 cells in the peritoneal cavity. Mice were euthanized 24 hr after injection, and B-1 cells were analyzed by flow cytometry. Additionally, mice received CFSE-labeled ApoB derived from 2 million initial thy-mocytes at 12 hr after injection. Mice were then analyzed for ApoB staining on subcapsular sinus macrophages in the spleen, as described above, by flow cytometry.

## Proteinuria

Protein concentration was determined in the urine by BCA assay (Thermo Fisher Scientific) according to the manufacturer's protocol.

## Phagocyte depletion and adoptive transfer

WT mice received 200 µL of Clophosome-A Anionic Clodronate Liposomes (FormuMax) or control liposomes by intravenous injection to deplete splenic phagocyte populations. Twenty-four hours

after injection, splenic B cells were isolated from Gal9KO mice by negative selection (EasySep, Stemcell) and labeled with eFluor 450 dye, according to the manufacturer's instructions (ThermoFisher). $2 \times 10^6$ Gal9KO B cells were transferred by IV injection into control or depleted hosts. Twenty-four hours after adoptive transfer spleens were collected, and Gal9 expression was measured on endogenous and transferred B cells by flow cytometry. Cells were stimulated with titrated concentrations of anti-IgM F(ab')$_2$, as above. Depletion of phagocyte populations was determined by flow cytometry.

## NZB/W F1 hybrid model of murine lupus

Gal9KO mice were backcrossed onto NZB and NZW strains for five generations. NZB and NZW mice heterozygous for the Gal9KO allele were crossed to generate NZB/W KO and WT littermate controls. Blood and urine collection was performed biweekly to monitor disease progression. Autoantibody production and proteinuria was quantified as described above.

## Code availability

Thunderstorm plugin for ImageJ is available at http://zitmen.github.io/thunderstorm/ (*Ovesny, 2021*). DBSCAN algorithm is available in the SMoLR package, and the Hopkins index in the Spatstat package in R, available through Github.

## Acknowledgements

We thank Joan Wither (Krembil Research Institute, University Health Network) for MD4/ML5 mice, Juan Carlos Zúñiga-Pflücker (Sunnybrook Research Institute) for OP9- R7FS cells, and Mario Ostrowski (University of Toronto) for rGal9. This work was supported by funding from the Canadian Institutes of Health Research (CIHR; MOP-136808 and PJT-165938), and Canada Research Chair (CRC; 905–231134) to BT.

## Additional information

### Competing interests

Bebhinn Treanor: BT is a founder of Radiant Biotherapeutics and is a member of its Scientific Advisory Board. The other authors declare that no competing interests exist.

### Funding

| Funder | Grant reference number | Author |
|---|---|---|
| Canadian Institutes of Health Research | MOP-13608 | Bebhinn Treanor |
| Canadian Institutes of Health Research | PJT-165938 | Bebhinn Treanor |
| Canada Research Chairs | 905-231134 | Bebhinn Treanor |

The funders had no role in study design, data collection and interpretation, or the decision to submit the work for publication.

### Author contributions

Logan K Smith, Conceptualization, Data curation, Formal analysis, Investigation, Visualization, Methodology, Writing - original draft, Writing - review and editing; Kareem Fawaz, Formal analysis, Investigation, Visualization, Writing - review and editing; Bebhinn Treanor, Conceptualization, Resources, Data curation, Supervision, Funding acquisition, Writing - original draft, Project administration, Writing - review and editing

### Author ORCIDs

Bebhinn Treanor (iD) https://orcid.org/0000-0002-8626-5944

## Ethics

Animal experimentation: All procedures were approved by the Local Animal Care Committee at the University of Toronto Scarborough, Animal Use Protocols 20012282 and 20011481.

## Decision letter and Author response

Decision letter https://doi.org/10.7554/eLife.64557.sa1
Author response https://doi.org/10.7554/eLife.64557.sa2

# Additional files

## Supplementary files

- Source data 1. Source data for figures and figure supplements.

- Transparent reporting form

## Data availability

Lgals9-/- mice require an MTA from The Scripps Research Institute (TSRI) and MD4/ML5 (B6.Ig/sHEL) mice require an MTA from University Health Network (UHN). Source data files for quantification of flow cytometry, imaging, and ELISA data are available as excel files accompanying the manuscript.

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

# Appendix 1

**Appendix 1—key resources table**

| Reagent type (species) or resource | Designation | Source or reference | Identifiers | Additional information |
|---|---|---|---|---|
| Strain, strain background (*Mus musculus*) | C57BL/6 | Charles River Laboratories | Strain code: 027 | |
| Strain, strain background (*Mus musculus*) | Gal9-/- | Steven Beverley (Washington University) on behalf of the Scripps Research Institute | Gal9KO | |
| Strain, strain background (*Mus musculus*) | Nr4a1-eGFP (Nur77-GFP) | Jackson Laboratories | Strain code: 016617 | |
| Strain, strain background (*Mus musculus*) | New Zealand Black (NZB/BINJ) | Jackson Laboratories | Strain code: 000684 | |
| Strain, strain background (*Mus musculus*) | New Zealand White (NZW/LacJ) | Jackson Laboratories | Strain code: 001058 | |
| Strain, strain background (*Mus musculus*) | MD4 | Jackson Laboratories | Strain code: 002595 | |
| Strain, strain background (*Mus musculus*) | μMT | Jackson Laboratories | Strain code: 002288 | |
| Strain, strain background (*Mus musculus*) | MD4/ML5 | Joan Wither (University of Toronto) | | |
| Cell Line (*Homo sapiens*) | Male:HEp2 | ATCC | Cat#: CCL-23 | |
| Cell Line (*Mus musculus*) | OP9-R7FS | Juan Carlos Zúñiga-Pflücker (University of Toronto) | | |
| Antibody | Phospho-Tyrosine (p-Tyr) | Sunnybrook Research Institute | | Clone:4G10 |
| Antibody | Anti-IgM F(ab')$_2$ | Jackson Immunoresearch | Cat#: 155-006-020 | Polyclonal |
| Antibody | Anti-B220-Pacific Blue | BioLegend | Cat#: 103227 | Clone: RA3-6B2 |
| Antibody | Anti-CD11b-Pacific Blue | BioLegend | Cat#: 101224 | Clone: M1/70 |
| Antibody | Anti-Ly6G-FITC | BioLegend | Cat#:127606 | Clone: 1A8 |
| Antibody | Anti-CD45- Brilliant Violet 510 | BioLegend | Cat#: 103138 | Clone: 30-F11 |
| Antibody | Anti-CD169-PE-Cyanine7 | BioLegend | Cat#: 142412 | Clone: 3D6.122 |
| Antibody | Anti-F4/80-Alexa Fluor 700 | BioLegend | Cat#: 123129 | Clone: BM8 |
| Antibody | Anti-CD11c-APC | BioLegend | Cat#: 117310 | Clone: N418 |
| Antibody | Anti-CD64-PE | BioLegend | Cat#: 139303 | Clone: X54-5/7.1 |
| Antibody | Anti-MHCII (I-A/I-E)-Brilliant Violet 605 | BioLegend | Cat#: 107639 | Clone: M5/114.15.2 |
| Antibody | Anti-CD8-APC | BioLegend | Cat#: 100712 | Clone: 53-6.7 |
| Antibody | Anti-CD4-Brilliant Violet 510 | BioLegend | Cat#: 100449 | Clone: GK1.5 |

*Continued on next page*

*Appendix 1—key resources table continued*

| Reagent type (species) or resource | Designation | Source or reference | Identifiers | Additional information |
|---|---|---|---|---|
| Antibody | Anti-CD23-APC | BioLegend | Cat#: 101620 | Clone: B3B4 |
| Antibody | Anti-CD21/35-APC-Cyanine7 | BioLegend | Cat#: 123418 | Clone: 7E9 |
| Antibody | Anti-Galectin-9-PE | BioLegend | Cat#: 137904 | Clone: 108A2 |
| Antibody | Anti-Galectin-9-Alexa Fluor 488 | BioLegend | Cat#: 137908 | Clone: 108A2 |
| Antibody | Anti-Galectin-9 | BioLegend | Cat#: 137902 | Clone: 108A2 |
| Antibody | Anti-CD86-FITC | BioLegend | Cat#: 105006 | Clone: GL-1 |
| Antibody | Anti-CD86-PE-Cyanine7 | BioLegend | Cat#: 105014 | Clone: GL-1 |
| Antibody | Anti-IgM-Alexa Fluor 488 | Jackson Immunoreasearch | Cat#: 715-547-020 | |
| Antibody | Anti-GL7-APC | BioLegend | Cat#: 144618 | Clone: GL7 |
| Antibody | Anti-CD95-FITC | BioLegend | Cat#: 152606 | Clone: SA367H8 |
| Antibody | Anti-IgD-Brilliant Violet 605 | BioLegend | Cat#: 405727 | Clone: 11-26c.2a |
| Antibody | Anti-CD138-Brilliant Violet 605 | BioLegend | Cat#: 142516 | Clone: 281-2 |
| Antibody | Anti-CD38-PE-Cyanine7 | BioLegend | Cat#: 102718 | Clone: 90 |
| Antibody | Anti-IgM-Alexa Fluor 647 | Jackson Immunoresearch | Cat#: 155-607-020 | |
| Antibody | Anti-IgG (FC)-Alexa Fluor 488 | Jackson Immunoresearch | Cat#: 155-545-008 | |
| Antibody | Anti-PD-1-PE-Cyanine7 | BioLegend | Cat#: 109110 | Clone: RMP1-30 |
| Antibody | Anti-CXCRV-Brilliant Violet 421 | BioLegend | Cat#: 145512 | Clone: L138D7 |
| Antibody | Anti-FoxP3-PE | BioLegend | Cat#: 126404 | Clone: MF-14 |
| Antibody | Anti-IgMa-PE | BioLegend | Cat#: 408608 | Clone: MA-69 |
| Antibody | Anti-B220-Alexa Fluor 647 | BioLegend | Cat#: 103226 | Clone: RA3-6B2 |
| Antibody | Anti-CD5-PE-Cyanine 5 | BioLegend | Cat#: 100610 | Clone: 53-7.3 |
| Antibody | Anti-CD19-Pacific Blue | BioLegend | Cat#: 115523 | Clone: 6D5 |
| Antibody | Anti-CD43-APC | BioLegend | Cat#: 143208 | Clone: S11 |
| Antibody | Anti-CD23-PE | BioLegend | Cat#: 101607 | Clone: B3B4 |
| Antibody | Anti-CD69-Brilliant Violet 510 | BioLegend | Cat#: 104531 | Clone: H1.2F3 |
| Antibody | Anti-CD5-Alexa Fluor 647 | BioLegend | Cat#: 100614 | Clone: 53-7.3 |
| Antibody | Anti-SiglecG-APC | Fisher Scientific | Cat#: 501123146 | Clone: SH2.1 |
| Antibody | Anti-CD22-Alexa Fluor 647 | BioLegend | Cat#: 126108 | Clone: OX-97 |
| Antibody | Anti-CD180 | BioLegend | Cat#:117710 | Clone:RP/14 |
| Antibody | Anti-TLR4-Alexa Fluor 488 | Fisher Scientific | Cat#: 5016829 | Clone: UT41 |
| Antibody | Anti-C3 | BioLegend | Cat#: 518106 | Clone: K13/16 |
| Recombinant Proteins | Recombinant Galectin-9 | R&D Systems | Cat#: 3535-GA-050 | |
| Chemical compound/drug | Calf Thymus DNA | ThermoFisher Scientific | Cat#: 15633019 | |
| Chemical compound/drug | Phosphorylcholine | Sigma Aldrich | Cat#: P0378 | |
| Chemical compound/drug | Hen Egg Lysozyme | Sigma Aldrich | Cat#: 10837059001 | |

*Continued on next page*

*Appendix 1—key resources table continued*

| Reagent type (species) or resource | Designation | Source or reference | Identifiers | Additional information |
|---|---|---|---|---|
| Chemical compound/drug | β-lactose | Sigma Aldrich | Cat#: L3750 | |
| Chemical compound/drug | Streptavidin-Alexa Fluor 647 | Thermo Fisher Scientific | Cat#: S21374 | |
| Chemical compound/drug | CFSE | Thermo Fisher Scientific | Cat#: C34554 | |
| Chemical compound/drug | Cell tracker eFluor 450 | Thermo Fisher Scientific | Cat#: 65-0842-85 | |
| Chemical compound/drug | Puromycin Dihydrochloride | Thermo Fisher Scientific | Cat#: A1113803 | |
| Chemical compound/drug | LPS | Invivogen | Cat#: tlrl-eklps | |
| Chemical compound/drug | Imiquimod | Invivogen | Cat#: tlrl-imq | |
| Chemical compound/drug | Zymosan | Invivogen | Cat#: tlrl-zyn | |
| Chemical compound/drug | CpG-ODN | Invivogen | Cat#: tlrl-1585 | |
| Chemical compound/drug | M2 Beads | Sigma Aldrich | Cat#: M8823 | |
| Commercial assay/kit | B cell Isolation kit | Stemcell Technologies | Cat#: 19854 | |
| Commercial assay/kit | Mouse IgM Quantitation set | Bethyl Laboratories | Cat#: E90-101 | |
| Commercial assay/kit | Mouse IgG Quantitation set | Bethyl Laboratories | Cat#: E90-131 | |
| Commercial assay/kit | Clophosome-A and control liposome kit | FormuMax | Cat#: F70101C-AC | |
| Commercial assay/kit | Fluorescence quantitation beads | Bangs Labs | Cat#: 817 | |
| Commercial assay/kit | BCA kit | Sigma Aldrich | Cat#: A53225 | |
| Software, algorithm | Flowjo v10.6 | Tree Star | | |
| Software, algorithm | ImageJ | NIH/LOCI | | |
| Software, algorithm | ThunderSTORM | Github | zitmen | |
| Software, algorithm | Rstudio | Rstudio PBC | | |
| Software, algorithm | SMoLR | Github | ErasmusOIC | |
| Software, algorithm | Spatstat | Github | spatstat | |

