## [Decision Letter]

**Acceptance summary:**

By using galectin 9-KO mouse, authors nicely showed that galectin-9 plays a critical role in setting the threshold of B cell activation. Furthermore, they demonstrated that galectin-9 regulates the interaction between BCR and TLRs with CD5 and CD180. These findings well explain why loss of galectin-9 drives autoimmune responses.

**Decision letter after peer review:**

Thank you for submitting your article "Galectin-9 regulates the threshold of B cell activation and autoimmunity" for consideration by *eLife*. Your article has been reviewed by 2 peer reviewers, one of whom is a member of our Board of Reviewing Editors, and the evaluation has been overseen by Carla Rothlin as the Senior Editor. The reviewers have opted to remain anonymous.

The reviewers have discussed the reviews with one another and the Reviewing Editor has drafted this decision to help you prepare a revised submission.

As the editors have judged that your manuscript is of interest, but as described below that additional experiments are required before it is published, we would like to draw your attention to changes in our revision policy that we have made in response to COVID-19 (https://elifesciences.org/articles/57162). First, because many researchers have temporarily lost access to the labs, we will give authors as much time as they need to submit revised manuscripts. We are also offering, if you choose, to post the manuscript to bioRxiv (if it is not already there) along with this decision letter and a formal designation that the manuscript is "in revision at eLife". Please let us know if you would like to pursue this option. (If your work is more suitable for medRxiv, you will need to post the preprint yourself, as the mechanisms for us to do so are still in development.)

Summary:

The authors studied the role of the soluble glycan-binding protein galectin-9 which can be produced by B cells and binds to B cell surface proteins such as membrane IgM and CD45. They used galectin-9 (Gal-9) KO mice to study the consequence of loss of this protein on B cell activation. They convincingly show that Gal-9 KO B cells react with higher BCR signaling and react to lower concentrations of antigens. This is true both for conventional B-2, as well as for B-1a cells. For B-1a cells not only BCR-induced signaling, but also TLR-signaling is enhanced. These studies on signaling, B cell activation and antigen internalisation are thoroughly done. For B-1a cells, direct binding of Gal-9 to CD5, TLR4 and CD180 is identified. Here some controls of other receptors and a discussion how this binding could happen would be useful. The authors also show that Gal-9 KO mice develop autoimmunity upon ageing or that in autoimmune models, the disease occurs more early. These parts are also generally done well. Overall it is an impressive amount of work with many interesting novel data.

Essential revisions:

It is important to clarify what is the cellular source of galectin-9 secretion in regard to development of autoimmunity in Gal9KO mice. Do B cells themselves secrete galectin-9 which binds to B cell surface to mediate signals, or is there any contribution of other cell types? Analysis of B cell-specific Gal9 KO mice (such as mixed BM chimera with B cell-deficient μMT mice) may help to address this question.

---

## [Author Response]

Essential revisions:It is important to clarify what is the cellular source of galectin-9 secretion in regard to development of autoimmunity in Gal9KO mice. Do B cells themselves secrete galectin-9 which binds to B cell surface to mediate signals, or is there any contribution of other cell types? Analysis of B cell-specific Gal9 KO mice (such as mixed BM chimera with B cell-deficient μMT mice) may help to address this question.

We agree that the source of Gal9 is important to address. In the revised manuscript we have included a series of adoptive transfer experiments to help elucidate the source of Gal9 on the B cell surface within the spleen. In Figure 1 —figure supplement 2 we show the expression of intracellular and extracellular Gal9 on a variety of splenic immune cell populations as well as a metric of Gal9 expression that considers population size. From these data we noted that macrophages and B cells appear to be likely sources of Gal9 in situ. To further elucidate this, we performed an adoptive transfer of Gal9 KO B cells into WT, macrophage-depleted, or B cell deficient μMT hosts and measured Gal9 expression and the effect of Gal9 reconstitution on the threshold of activation. From this we conclude that both macrophages and B cells contribute to exogenous sources of Gal9 and that these are sufficient to alter the threshold of B cell activation.